# Distinct immune responses in people living with HIV following SARS-CoV-2 recovery

Dieter Mielke [1,2,7] ✉, Shuying Sue Li[3,7], Daniel J. Schuster [1,2,4], Xiaohong Li [3], Jiani Hu[3], Shelly Karuna [3], Kelly E. Seaton [1,2], Caroline Brackett[1,2], Brooke Dunn[2], Taylor Keyes[2], Adam Zalaquett[2], Sherry Stanfield-Oakley[2], Lu Zhang[1,2], Martina S. Wesley [1,2], Nathan Eisel[1,2], Nicole L. Yates[1,2], Xiaoying Shen [1,2], Lakshmanane Premkumar[5], Russell St. Germain[3], Anton M. Sholukh [3], Kristen Cohen[3], Stephen de Rosa[3], April Kaur Randhawa [3], John A. Hural[3], Lawrence Corey [3], M. Julianna McElrath [3], Georgia D. Tomaras[1,2,4,6], Ollivier Hyrien[3,8] & Guido Ferrari [1,2,6,8] ✉

## Abstract

**Background** SARS-CoV-2 infection results in greater disease severity among immunocompromised individuals compared to healthy individuals. However, there is conflicting information about the impact of chronic HIV infection on immune responses to SARS-CoV-2 infection and vaccination.

**Method** We used a combination of machine learning approaches and network analysis to explore 56 immune markers and comprehensively profile humoral and cellular immunity in a cross-sectional observational cohort of people without HIV (PWOH; $n = 216$) and people living with HIV (PLWH; $n = 43$) who recovered from SARS-CoV-2 infection (13–131 days since SARS-COV-2 diagnosis) early in the pandemic.

**Results** PLWH recovered from symptomatic outpatient COVID-19 exhibit lower humoral and B cell responses to SARS-CoV-2 vs. PWOH but, surprisingly, both symptomatic outpatient and hospitalized PLWH have higher anti-endemic coronavirus antibody responses compared to PWOH counterparts and asymptomatic PLWH. The latter observation suggests that this was not strictly due to broadly elevated levels of anti-endemic coronavirus antibodies in PLWH. Moreover, correlation-based analysis reveals that while different compartments of the immune response to SARS-CoV-2 infection are positively correlated in PWOH recovered from symptomatic outpatient COVID-19, these correlations are weaker in PLWH.

**Conclusion** Our analyses reveal significant differences in the coordinated immune responses elicited by infection in PLWH compared to PWOH.

## Plain language summary

COVID-19 tends to be severe in individuals with underlying health conditions compared to those without. In particular, it is unclear whether individuals living with HIV exhibit immune responses to SARS-CoV-2 that differ from those without HIV. To investigate this gap, we use tools to analyze and compare immune responses to SARS-CoV-2 in individuals with and without HIV, who had recovered from the infection. Our study reveals that people living with HIV exhibit weaker immune responses to SARS-CoV-2 but stronger responses to endemic coronaviruses. Additionally, we observed that immune readouts measured in people living with HIV show weaker correlations with each other compared to those in individuals without HIV. These findings suggest that chronic, treated HIV infection may influence the immune response to SARS-CoV-2.

The novel coronavirus, severe acute respiratory syndrome coronavirus 2 (SARS-CoV-2), the causative agent of coronavirus disease 2019 (COVID-19), has had a devastating effect since its detection in late 2019, resulting in over 6.8 million deaths globally[1]. In addition, despite multiple licensed vaccines that can effectively prevent severe COVID-19, SARS-CoV-2 infection continues to result in hospitalization and death, with higher rates among people who have not been vaccinated.

SARS-CoV-2 has been shown to result in greater disease severity among immunocompromised individuals due to immunologic deficiencies

[1]Center for Human Systems Immunology, Duke University, Durham, NC, USA. [2]Department of Surgery, Duke University, Durham, NC, USA. [3]Vaccine and Infectious Disease Division, Fred Hutchinson Cancer Center, Seattle, WA, USA. [4]Department of Integrative Immunobiology, Duke University School of Medicine, Durham, NC, USA. [5]Department of Microbiology and Immunology, University of North Carolina, Chapel Hill, NC, USA. [6]Department of Molecular Genetics and Microbiology, Duke University School of Medicine, Durham, NC, USA. [7]These authors contributed equally: Dieter Mielke, Shuying Sue Li. [8]These authors jointly supervised this work: Ollivier Hyrien, Guido Ferrari. ✉e-mail: dieter.mielke@duke.edu; gflmp@duke.edu

that limit effective prevention, treatment, and clearance of infection (reviewed in DeWolf et al.[2]). This group includes people living with HIV (PLWH). Numerous studies have associated increased COVID-19 severity and higher mortality rates with low CD4 counts in PLWH, likely a result of uncontrolled HIV infection[3-9]. In contrast, several studies of immune responses elicited by SARS-CoV-2 vaccination in PLWH with well-controlled HIV infection suggested comparable response magnitudes and short-term durability of both binding and neutralizing antibodies[10-13]. Consequently, the impact of HIV infection on SARS-CoV-2 immunity and vaccination response is still unclear.

Many studies investigating immune responses to SARS-CoV-2 infection or vaccination have focused on either antibody levels and functions or T cell responses. One study of both humoral and cellular responses in a small cohort of PLWH and PWOH who experienced predominantly mild COVID-19 has been conducted to date, finding relatively comparable responses in both groups 5–7 months post-infection 14.

In this study, we perform a comprehensive analysis of both humoral and cellular immune responses measured using a range of assays in a cross-sectional observational cohort of PWOH and PLWH who recovered from different levels of COVID-19 severity (13–131 days post-symptom onset) early in the pandemic. We use a combination of machine learning approaches and network analysis to explore the interrelation between immune readouts and determine how immune profiles and coordination of immune responses differed between PWOH and PLWH who had recovered from SARS-CoV-2 infection. Significant differences in both the magnitudes and coordination of SARS-CoV-2-specific and anti-endemic CoV immune responses are observed, particularly among participants recovered from symptomatic outpatient COVID-19. In this group, a subset of markers, including Spike-specific B cell frequencies and anti-endemic CoV binding antibodies, are able to classify PLWH from PWOH with a high degree of accuracy. These findings may inform the evaluation strategy of vaccine efficacy in these clinical groups.

## Methods

### Study approval

Institutional Review Board (IRB) approval was granted by a Central IRB (Advarra IRB) and, as applicable, by individual clinical research sites' IRBs (Supplementary Table 11). All participants provided written informed consent prior to participation.

### Study conduct and clinical trial information

Details of study conduct and clinical trial information were previously reported in Karuna, et al.[14]. Participants recovered from SARS-CoV-2 infection were enrolled between May and October 2020 in the HVTN 405/HPTN 1901 observational cohort study (NCT04403880) led by the COVID-19 Prevention Trials Network (CoVPN). US (n = 195) and Peruvian (n = 178) participants, including 43 PLWH, were stratified by peak symptom severity (asymptomatic, symptomatic outpatient, and hospitalized) and by age (18–55 and 55+ years of age). The applicable eligibility criterion for inclusion was: resolution of COVID-19 within 1–8 weeks of enrollment or, if asymptomatic infection, reports positive SARS-CoV-2 test within 2–10 weeks of enrollment. If participants had been hospitalized, they were not enrolled until after hospital discharge. Peak symptom severities were self-reported as asymptomatic if no symptoms were present at the time of diagnosis through recovery, symptomatic outpatient if any symptoms were reported but the participant was not hospitalized for COVID-19, and hospitalized if hospitalized due to COVID-19. It should be noted that in the early pandemic setting when this cohort was established (pre-vaccine, hospital systems crashing under the weight of the number of infections, variable adherence to other public health measures, shortages of workforce, personal protection equipment and other hospital resources etc.), a participant's admission was a physician judgment decision made in the context of the triage that naturally occurred under those circumstance when limited hospital resources were available and varied across and within regions as COVID-19 prevalence and severity varied over time. Therefore,

hospitalization was a reasonable proxy for disease severity, and no single consistent parameter was used. Detailed information on demographics and comorbidities was collected at the time of enrollment, along with self-reported date of positive direct viral detection testing (i.e., antigen or molecular test). HIV-1 status, CD4 counts, and HIV-1 viral loads were reported by the enrolling clinics from participants' health records. This study included samples only from the enrollment visit. All laboratories were blinded to HIV status until after analyses were complete, and all assays were conducted in compliance with Good Clinical Laboratory Practice guidelines for consistency and reproducibility.

### Binding antibody multiplex assays

SARS-CoV-2-specific and endemic CoV-specific IgG1, IgG3, and IgA were measured by Binding Antibody Multiplex Assay[15-17] with modifications. Briefly, antigens were bound to NeutrAvidin-coupled fluorescent microspheres (MagPlex, Luminex Corp, Austin, TX) via a biotinylated rabbit anti-6x His-tag antibody to directionally orient the F'(ab) arms outward. Prepared microspheres were incubated with human sera (IgG1 at 1:50, 1:1000, 1:10 000, 1:25 000; IgG3 and IgA at 1:50 and 1:250) and controls diluted in assay diluent for 2 h, shaking at 750 RPM and 22 °C. Subsequently, a mouse anti-human IgG1 (BioLegend, San Diego, CA; clone# 12G8G11) or IgG3 (Invitrogen, Waltham, MA; clone # HP6047) followed by goat anti mouse IgG-PE (SouthernBiotech, Birmingham, AL; catalog # 1030-09) were used to detect bound IgG1 and IgG3, respectively. Goat anti-human IgA-PE (Jackson ImmunoResearch, West Grove, PA; catalog # 109-006-011) was utilized to detect IgA. IgA samples were IgG-depleted prior to testing using a protein G MultiTrapTM plate (GE Healthcare Bio-Sciences AB, Uppsala, Sweden). Assay plates were read using a Bio-Plex 200 System (Bio-Rad, Hercules, CA). Sixty-eight SARS-CoV-2-seronegative samples, collected prior to Nov 2019, were tested at a 1:50 dilution to establish isotype- and antigen-specific positivity cut-offs (95th percentile and ≥100 net MFI) (BioIVT, Westbury, NY). Antigen panel components are listed in Supplementary Table 11. All samples, controls, and standards were assayed in duplicate, and the mean value reported. Negative controls and uncoupled microspheres were included in each assay to ensure specificity. Levey–Jennings charts were used to track antigen performance across assays. Response calls for SARS-CoV-2-specific antibodies were made with serum at a 1:50 dilution to increase sensitivity, while response magnitudes were reported at 1:1000 for IgG1 to increase the number of samples within the linear range. No response calls were made for the endemic CoV antigens.

### MSD Four-Plex SARS-CoV-2 IgG binding assay

SARS-CoV-2 Spike-, S1 RBD-, and nucleocapsid-specific IgG in serum samples were quantitatively measured using the V-PLEX SARS-CoV-2 384 Panel 1 (IgG) kit[18], according to manufacturer's instructions (Meso Scale Discovery (MSD), 394 Rockville, MD). Briefly, pre-coated MULTI-SPOT 384-Well plates were blocked (Blocker A solution) for 1 h at 20–26 °C. Plates were washed with MSD Wash Buffer, and samples were added to the plate, tested in duplicate at 1:500, 1:10,000, 1:200,000, and 1:4,000,000 dilutions. Plates were washed after 4 h, and binding was detected using a mouse anti-human IgG conjugated to MSD SULFO-TAGTM. Following the addition of MSD GOLDTM Read Buffer B, plates were read on a MESO SECTOR S 600MM instrument. Sixty-six SARS-CoV-2-seronegative serum samples were tested at a 1:500 dilution and processed to establish the positivity cut-off (mean plus 3 standard deviations) (BioIVT, Westbury, NY). The magnitude of binding in arbitrary units per milliliter (AU/mL) was calculated at each sample dilution by backfitting to a 7-place calibration curve run in duplicate on each plate. The median AU/mL from all dilutions in the linear range of the curve was used to calculate the final AU/mL for each sample. Conversion to WHO/NIBSC International Standard Units of Binding Antibody Units (BAU/mL) was calculated with MSD units (AU/mL) × a conversion factor for Reference Standard 1 (Lot A00V004) (0.00236, 0.0272, 0.00901 for nucleocapsid, RBD, and Spike, respectively) available through MSD.

## Spike protein-expressing cell antibody binding assay

The Spike-transfected cell antibody binding assay was used to measure antibody binding to Spike on the surface of transfected cells[19]. Target cells were derived by transfection with plasmids designed to express the SARS-CoV-2 D614 Spike protein with a c-terminus flag tag. Cells not transfected with any plasmid (mock transfected) were used as a negative control condition. After resuspension, washing, and counting, $1 \times 10^5$ Spike-transfected target cells were dispensed into 96-well V-bottom plates and incubated with six serial dilutions of human plasma from infected participants starting at a 1:50 dilution. Mock-transfected cells were used as a negative control. After 30 min incubation at 37 °C, cells were washed twice with 250 μL/well of PBS, stained with vital dye (Live/Dead Far Red Dead Cell Stain, Invitrogen) to exclude nonviable cells from subsequent analysis, washed with Wash Buffer (1%FBS-PBS; WB), permeabilized with CytoFix/CytoPerm (BD Biosciences), and stained with 1.25 μg/mL anti-human IgG Fc-PE/Cy7 (Clone HP6017; Biolegend) and 5 μg/mL anti-flag-FITC (clone M2; Sigma Aldrich) in the dark for 20 min at room temperature. After three washes with Perm Wash (BD Biosciences), the cells were resuspended in 125 μL PBS-1% paraformaldehyde. Samples were acquired within 24 h using a BD Fortessa cytometer and a High Throughput Sampler (HTS, BD Biosciences). Data analysis was performed using FlowJo 10 software (BD Biosciences). A minimum of 50,000 total events was acquired for each analysis. Gates were set to include singlet, live, flag+ (Spike+), and IgG+ events. Binding to mock-infected cells was measured using the live cell gate, as there were no flag+ events. For example, gates, see Supplementary Fig. 7. All final data represent specific binding, determined by subtraction of non-specific binding observed in assays performed with mock-transfected cells.

## Neutralization assay

Neutralizing antibody ID50 and ID80 values of all samples were measured by the VSV pseudovirus neutralization assay[20]. Vero cells were seeded at $2 \times 10^4$ cells/well in black-walled 96-well plates 24 h before the assay was performed. A 7-point, 3-fold dilution curve was generated with a starting sample dilution of 1:20. PsVSV-Luc-D19 ($3.8 \times 10^2$ TCID50) were mixed with the plasma dilutions, incubated at 37 °C in 5% $CO_2$ for 30 min, and then transferred onto Vero cells. Cells were incubated for 18–20 h. Luciferase activity was measured by Bio-Glo Luciferase Assay System (Promega, Madison, WI) using a 2030 VICTOR X3 multilabel reader (PerkinElmer, Waltham, MA). Percent neutralization was calculated by the following equation: [1−(RLU with sample−RLU with uninfected cells)/(RLU with infected cells−RLU with uninfected cells)] × 100. Plasma from a subject with severe, PCR-confirmed SARS-CoV-2 infection collected following discharge from the hospital was used as a positive control. Pooled human serum collected in 2015-2018 was used as a negative control.

VSV pseudovirus (PsVSV-Luc-D19) were prepared using a codon-optimized gene of SARS-CoV-2 spike protein (YP_009724390.1) cloned into pcDNA3.1, provided by Dr. F.A. Lempp (Vir Biotechnologies), and VSV(G*ΔG-luciferase) system purchased from Kerafast (Boston, MA)[21,22]. VSV(G*ΔG-luciferase) pseudotyped with SARS-CoV-2 spike was produced in 293T cells and stored at −80 °C. Median tissue culture infectious dose (TCID50) was measured using Vero cells (catalog number CCL-81; ATCC) with serial 2-fold dilutions of the prepared pseudovirus.

Positivity calls were based on whether or not any neutralization was observed through the entire titration (starting dilution 1:20) with maximum percent inhibition (MPI) between 10%–50%. For the positive calls where the neutralization did not reach 50% or 80%, ID50/ID80 titer was first estimated by Prism v. 9.3 (GraphPad, San Diego, CA) using a four-parameter logistic curve. If the estimation failed, then the ID50/ID80 titer was set to be 10. For negative calls (MPI < 10%), the ID50/ID80 titer was set to be 5 for graphing purposes.

## MSD ACE2 blocking assay

Antibodies that block binding of SARS-CoV-2 Spike to ACE2 were quantitatively measured using the V-PLEX SARS-CoV-2 Panel 2 (ACE2) kit according to the manufacturer's instructions (MSD, Rockville, MD).

Briefly, SARS-CoV-2 spike-coated MULTI-SPOT 96-Well plates were blocked and washed as above. Samples were tested in duplicate at a dilution of 1:250. 415 Samples were selected based on RBD-specific IgG1 response magnitudes in a semi-random way using the following approach: (1) samples with positive responses passing quality control were evenly divided into top, middle, and bottom thirds and high blank (blank MFI > 5000), (2) random numbers were assigned, and (3) 25 samples were selected from each tertile along with all 24 from the high blank group. Sixty-eight additional samples (blinded to our lab) were added to include all PLWH samples. A 7-place calibration curve and blank well were run in duplicate on each plate, as well as a positive control mutant ACE2 protein (4-fold, 4-place dilution starting at 6 μg/mL). Samples were incubated with human ACE2 protein conjugated to MSD SULFO-TAGTM, washed, and read as above. Seventy-two SARS-CoV-2-seronegative samples were tested at a 1:250 dilution and processed to establish the positivity cut-off (mean plus 3 standard deviations, after truncating all negative values to zero). Percent blocking for samples was calculated from the 7-place calibration curve using the following equation: (1 − (Sample electrochemiluminescent (ECL) Signal Mean − Calibrator 1 ECL Signal Mean)/(Blank well ECL Signal Mean − Calibrator 1 ECL Signal Mean)) × 100.

## Antibody-dependent NK cell degranulation assays (infected and Spike-transfected)

Cell-surface expression of CD107a was used as a marker for NK cell degranulation, a prerequisite process for, and strong correlate of, ADCC[23] was measured[19]. Target cells were either Vero E6 cells after a 2-day infection with SARS-CoV-2 USA-WA1/2020 or 293T cells 2 days post-transfection with a SARS-CoV-2 S protein (D614) expression plasmid. Natural killer cells were purified by negative selection (Miltenyi Biotech) from peripheral blood mononuclear cells obtained by leukapheresis from a healthy, SARS-CoV-2-seronegative individual (Fc-gamma-receptor IIIA (FcγRIIIA)158 V/F heterozygous) and previously assessed for FcγRIIIA genotype and frequency of NK cells were used as a source of effector cells. NK cells were incubated with target cells at a 1:1 ratio in the presence of diluted plasma or monoclonal antibodies, Brefeldin A (GolgiPlug, 1 μL/mL, BD Biosciences), monensin (GolgiStop, 4 μL/6 mL, BD Biosciences), and anti-CD107a-FITC (BD Biosciences, clone H4A3) in 96-well flat bottom plates for 6 h at 37 °C in a humidified 5% $CO_2$ incubator. NK cells were then recovered and stained for viability prior to staining with CD56-PECy7 (BD Biosciences, clone NCAM16.2), CD16-PacBlue (BD Biosciences, clone 3G8), and CD69-BV785 (Biolegend, Clone FN50). The NK cells from the infected cell assay were incubated with 0.2 mL of 4% Methanol-free Formaldehyde (Duke GHRB SOP 38; Attachment 17) for 30 min at Room Temperature. Cells were then resuspended in 115 μL PBS-1% paraformaldehyde. Flow cytometry data analysis was performed using FlowJo software (v10.8.0). Data is reported as the % of CD107a+ live NK cells (gates included singlets, lymphocytes, aqua blue−, CD56+, and/or CD16+, CD107a+). For example, gates, see Supplementary Fig. 8. All final data represent specific activity, determined by subtraction of non-specific activity observed in assays performed with mock-infected cells and in the absence of antibodies.

## Antibody-dependent cellular phagocytosis (ADCP)

The ADCP assay was modeled after prior work[24,25] with modifications. Quantification of ADCP was performed by covalently binding 6P Spike (HexaPro)[26] to NeutrAvidin fluorescent beads (ThermoFisher, Waltham, MA) and forming immune complexes by incubation with 1:50 diluted serum. This dilution was chosen from a 6-place 5-fold titration series starting from 1:10. HexaPro was used based on its more highly stabilized trimer conformation than 2P spike[26]. Monoclonal antibodies CV23 IgG1 and CV30 IgG1[27], and CR3022 IgG1 served as positive controls, while CH65 IgG1 served as a negative control[28]. Immune complexes were incubated with THP-1 cells (ATCC, Manassas, VA), and cellular fluorescence was measured using a BD LSR Fortessa (BD Biosciences, San Jose, CA). For example, gates, see Supplementary Fig. 9. Seventy-two SARS-CoV-2-seronegative

samples were tested at a 1:50 dilution and processed to establish the positivity cut-off (95th percentile and 3 times the median) (BioIVT, Westbury, NY). ADCP scores were calculated as (mean fluorescence intensity (MFI) × frequency of phagocytosis-positive cells)/(MFI × frequency of bead-positive cells in a PBS control well).

## B cell phenotyping assay

Fluorescent SARS-CoV-2-specific S6P[29] (provided by Roland Strong, Fred Hutchinson Cancer Research Center, Seattle, WA) and RBD (provided by Leonidas Stamatatos, Fred Hutchinson Cancer Research Center, Seattle, WA) probes were made by combining biotinylated protein with fluorescently labeled streptavidin (SA). The S6P probes were made at a ratio of 1:1 molar ratio of trimer to SA. Two S6P probes, one labeled with Alexa-Fluor488 (Invitrogen), one labeled with AlexaFluor647 (Invitrogen), were used in this panel to increase the specificity of the detection of SARS-CoV-2-specific B cells. The RBD probe was prepared at a 4:1 molar ratio of RBD monomers to SA, labeled with R-phycoerythrin (Invitrogen). Cryopreserved PBMCs from SARS-CoV-2-convalescent participants and a pre-pandemic SARS-CoV-2-naive donor were thawed at 37 °C and stained for SARS-CoV-2-specific memory B cells as described previously[27] with a panel of fluorescently labeled antibodies (Supplementary Table 12). Cells were stained first with the viability stain (Invitrogen) in PBS for 15 min at 4 °C. Cells were then washed with 2% FBS/PBS and stained with a cocktail of the three probes for 30 min at 4 °C. The probe cocktail was washed off with 2% FBS/PBS, and the samples were stained with the remaining antibody panel and incubated for 25 min at 4 °C. The cells were washed two times and resuspended in 1% paraformaldehyde/PBS for collection on an LSR II or FACSymphony flow cytometer (BD Biosciences). Data were analyzed in Flow Jo version 9.9.4. Examples of the gates are provided in Supplementary Fig. 10.

## Intracellular cytokine staining assay

Flow cytometry was used to examine SARS-CoV-2-specific CD4+ and CD8+ T cell responses using a validated ICS assay. The assay was similar to published reports[30–32] and the details of the staining panel are included in Supplementary Table 13. Peptide pools covering the structural proteins of SARS-CoV-2 were used for the six-h stimulation. Peptides matching the SARS-CoV-2 Spike sequence (316 peptides, plus 4 peptides covering the G614 variant) were synthesized as 15 amino acids long with 11 amino acids overlap and pooled in 2 pools (S1 and S2) for testing (BioSynthesis). All other peptides were 13 amino acids overlapping by 11 amino acids, and were synthesized by GenScript. The peptides covering the envelope (E), membrane (M) and nucleocapsid (N) were initially combined into one peptide pool, but most of the assays were performed using a separate pool for N and one that combined only E and M. Several of the open reading frame (ORF) peptides were combined into two pools: ORF 3a and 6, and ORF 7a, 7b and 8. All peptide pools were used at a final concentration of 1 mg/mL for each peptide. As a negative control, cells were not stimulated; only the peptide diluent (DMSO) was included. As a positive control, cells were stimulated with a polyclonal stimulant, staphylococcal enterotoxin B. For example gates, see Supplementary Fig. 11. Cells expressing IFN-g and/or IL-2 and/or CD154 were the primary immunogenicity endpoint for CD4+ T cells, and cells expressing IFN-g were the primary immunogenicity endpoint for CD8+ T cells. The overall response to SARS-CoV-2 was defined as the sum of the background-subtracted responses to each of the individual pools. A sample was considered positive for CD4+ or CD8+ T cell responses to SARS-CoV-2 if any of the CD4+ or CD8+ T cell responses to the individual peptide pool stimulations were positive. Positivity was determined using MIMOSA[33]. The total number of CD4+ T cells must have exceeded 10,000, and the total number of CD8+ T cells must have exceeded 5000 for the assay data to be included in the analysis.

## Statistics and reproducibility

Participant characteristics were compared between PLWH and PWOH using chi-square test for categorical variables and a two-sided t-test for continuous variables.

## Association of HIV status with immune markers adjusting for confounders

For each immune marker, response rate (where positivity cut-offs had been previously established for the assay) and magnitudes among all or positive responders (where positivity cut-offs had been previously established for the assay) between PLWH and PWOH were compared using the Firth logistic[34] and log-linear regressions, adjusting for all potential confounders (SARS-CoV-2 infection severity, age, sex assigned at birth, region (US vs Peru), current cigarettes or marijuana smoking status, and days since SARS-CoV-2 diagnosis). The comparisons were further carried out between PLWH and PWOH stratifying by SARS-CoV-2 infection severity using the regression models described above, plus an interaction between HIV-1 serostatus and SARS-CoV-2 infection severity and appropriate contrasts. Q values were calculated adjusting for multiple comparisons involving multiple antigens in each type of response measures using the Benjamini and Hochberg method to control the false discovery rate (FDR)[35]. P values ≤ 0.05 and q values ≤ 0.2 were used to indicate significance. A q value of 0.2 was used to be less restrictive and generate hypotheses for future studies.

## Immune markers normalization

First, we deselected any immune markers that were highly correlated with other immune markers (Spearman's correlation > 0.95); performed the log-transformation for binding neutralizing antibodies, B cells and T cells markers; imputed missing observations using Lasso linear regression method and the mice R package; and eliminated the effect of days since SARS-CoV-2 diagnosis on immune markers observations via the linear regressing the marker observations on days since SARS-CoV-2 diagnosis ($y \sim \alpha + \beta x$) and then subtracting the effect from the marker observations ($y - \beta x$). Then we normalized the markers with a mean of zero and a standard deviation of one.

## Polar plots

First, for each immune marker, the quantiles of the individuals' immune responses were calculated by ranking the individuals' immune responses and scaling the ranks from 0 to 1 among all participants. Then the mean quantile was calculated by HIV status overall or stratified by SARS-CoV-2 infection severity. Polar plots displayed the mean quantiles of all immune markers by HIV status overall or stratified by SARS-CoV-2 infection severity. The direction-projection-permutation test was used to assess whether the distribution of all immune markers differed globally between PLWH and PWOH, overall and stratified by SARS-CoV-2 infection severity.

## Correlation plots

Spearman correlations between immune markers were displayed in numbers at the lower triangle and circles at the upper triangle using the corrplot R package. The size of circles and the color of numbers and circles indicate the magnitude and direction (blue for positive and red for negative) of the correlations. The significant correlations are marked in white within the circles (* for $p < 0.05$, ** for $p < 0.01$, *** for $p < 0.001$) based on the exact two-sided test. Spearman correlations were evaluated by HIV status overall or stratified by COVID-19 severity.

## Network correlation analysis

Association between pairs of immune markers was measured using Spearman's rank correlation coefficients, by HIV status and SARS-CoV-2 infection severity. P values assessing the significance of these coefficients were computed and adjusted for multiple comparisons using the Benjamini-Hochberg procedure to correct the FDR. The correlation matrix was visualized as a graph in which an edge between two markers was retained if the associated correlation coefficient was significant (FDR < 10%) and greater than 0.70 in absolute value. Immune markers were subsequently clustered based on edge betweenness in the graph. Correlation coefficients among PWOH and PLWH were compared to each other within each COVID-19 infection severity. The sets of significant pairwise associations

(one per infection level category) were visualized using a network with a circular layout.

The number of high (correlation coefficient with absolute value > 0.7) and significant (FDR < 0.1) correlation coefficients, the number of non-zero correlations (size), and the average number of non-zero correlations per marker with the other markers (average degree) were used to measure properties of correlation networks. The network size, when defined as the number of non-zero correlation coefficients, was estimated by $(1-\pi_0)$ times the number of all possible pairs among 56 immune markers (1540), where $\pi_0$ was estimated using the $q$ value R package[36]. The average degree of the network was defined as two times the size of the network divided by the number of markers.

To assess the impact of the difference in sample size between PWOH and PLWH overall or within the COVID-19 severity group on estimates of network properties (size, average degree), we generated 10,000 random subsamples of observations from PWOH, each having the same sample size as their PLWH counterparts.

## Classification of PLWH vs PWOH

To avoid overfitting and identify a minimal number of the most predictive immune markers, we first applied the recursive feature-elimination algorithm with resampling and the random forest model to down-select markers to be included in the classifier[37,38]. The 10-fold cross-validation random forest method was used to select the best set of markers. This process was repeated 200 times to achieve a robust feature (immune marker) selection. We selected the markers that appeared in the best sets of markers, more than 90–95% out of 200 repeats. If more than 20 immune markers were selected, we would repeat this down-selection process among the selected markers. Secondly, we applied a backward selection procedure to further evaluate/eliminate any redundant features using 10-fold cross-validation and 30 repeats. Super Learner with three prediction algorithms (SL.randomForest, SL.svm, SL.extraTrees) and the mean cross-validated area under the receiver operating characteristic curves (CV-AUC) as loss function. Finally, the ROC curves and mean CV-AUC with 95% CI of the classifier using the final set of most predictive markers were constructed.

All analyses were performed using R[39].

## Reproducibility

All experimental assays were conducted using good clinical laboratory practices with rigorous pass/fail criteria and established cut-off values for positivity. All analyses were conducted on a total of $n = 216$ PWOH and $n = 43$ PLWH.

## Reporting summary

Further information on research design is available in the Nature Portfolio Reporting Summary linked to this article.

## Results

### Participant characteristics

We conducted a comprehensive analysis of immune responses using serum samples collected at the enrollment visit from a subset of participants enrolled in the HVTN 405/HPTN 1901 trial: a US- and Peru-based study conducted between May and October 2020[14]. The analysis comprised 43 samples from PLWH and 216 samples from PWOH, selected to match key demographic characteristics of the PLWH group.

PLWH and PWOH groups exhibited comparable median age, median time from SARS-CoV-2 diagnosis to enrollment, distribution of COVID-19 severity, and medical comorbidities (Table 1). However, the proportion of individuals assigned male at birth was significantly higher among PLWH compared to PWOH (83.7% vs. 50%, $p < 0.001$). There was also a higher percentage of black non-Hispanic race/ethnicity individuals in the PLWH group compared to PWOH (30.2% vs. 10.2%, $p = 0.004$), and a greater percentage of PLWH reporting active cigarettes or marijuana smoking (30.2% vs. 8.8%, $p < 0.001$) (Table 1). The proportion of PLWH was slightly higher in the US than in Peru (19.1% vs. 8.6%, $p = 0.055$). Among the 43

PLWH, 42 reported taking antiretroviral therapy (ART), and 24 out of 27 (85.2%) had viral loads below 50 copies/mL. CD4 counts at the time of sample collection were measured in 26 PLWH; counts exceeded 300 cells/μL in 24 (92.3%) of them[17].

## Differences in immune profiles between PLWH and PWOH

We evaluated humoral and cellular immune responses using eleven assays that measured a total of 56 unique immune markers against multiple SARS-CoV-2 and endemic CoV antigens, encompassing a broad range of antibody functions and cellular phenotypes (Supplementary Fig. 1). The assays included binding antibody assays quantifying total IgG, IgG1, IgG3, and IgA; functional antibody assays (neutralization, ACE2 blocking, antibody-dependent cellular phagocytosis, and antibody-dependent cellular cytotoxicity); B cell phenotyping of total and memory IgM-, IgA- and IgG-expressing B cells; and cellular assays quantifying activated CD4 and CD8 T cells expressing IFN-γ and/or IL-2.

We first explored overall immune profile differences between the complete set of PWOH and PLWH, as well as differences stratified by disease severity (Figs. 1a, b and S2). The direction-projection-permutation test showed that the global distribution of these immune markers differed significantly between PLWH and PWOH among all participants ($p = 0.014$, Fig. 1a) and among participants who recovered from symptomatic outpatient COVID-19 ($p = 0.025$; Fig. 1b) but not among those who recovered from either asymptomatic infection ($p = 0.93$) or hospitalization ($p = 0.14$) (Supplementary Fig. 2). Overall, SARS-CoV-2-specific immune responses tended to be lower and endemic-CoV-specific binding antibody responses tended to be higher in PLWH compared to PWOH (Fig. 1a).

Among participants who recovered from symptomatic outpatient COVID-19, the mean quantiles of immune markers in PWOH were mostly close to the overall mean quantiles of responses (0.5 indicated by a red circle in Fig. 1b), while responses in PLWH were more variable. Multiple markers in PLWH tracked lower than the overall mean quantile, including %Spike+ IgG+ total and %Spike+ and RBD+ memory B cells, while others were higher than the overall mean quantile, such as IgG1 binding antibodies to HKU1, NL63, and OC43 RBDs (Fig. 1c).

## Covariates-adjusted analyses reveal significant differences in immune marker magnitudes between PWOH and PLWH

Because differences in immune responses between PLWH and PWOH may be confounded by demographic factors and other variables, we next explored whether any significant differences existed for each of the 56 markers individually between PWOH and PLWH after adjusting for age, sex assigned at birth, current cigarettes or marijuana smoking status, region, and days since SARS-CoV-2 diagnosis at enrollment and stratifying by COVID-19 severity. While response rates did not significantly differ between PLWH and PWOH for any of the measured immune markers (Supplementary Tables 1–6, Supplementary Data 2 and 3), 13 markers exhibited significant differences in magnitudes ($p$ values ≤ 0.05 and $q$ values ≤ 0.20; Table 2). Eight of these significant differences were observed in the symptomatic outpatient group comparisons, five were from hospitalization group comparisons, and there were no statistically significant differences in the analysis restricted to asymptomatic participants (Table 2).

In participants who recovered from symptomatic outpatient COVID-19, all eight significantly different markers measured humoral responses. Seven of the markers were SARS-CoV-2-specific and significantly lower in PLWH than PWOH: IgG1 binding to SARS-CoV-2 Receptor Binding Domain (RBD), Nucleoprotein, N Terminal Domain (NTD), and 6-proline Spike (6P Spike); IgG binding to Spike and RBD; and Neutralization (ID50). The remaining marker was anti-endemic-CoV (IgG1 binding to OC43 RBD) and was significantly higher in PLWH (Table 2).

In participants who had recovered from hospitalization, all five immune markers significantly different between PWOH and PLWH were

**Table 1 | Participants' characteristics at enrollment and compared between people living with HIV (PLWH) and people without HIV (PWOH)**

| Characteristics | Levels | PWOH (n = 216) | PLWH (n = 43) | P value |
|---|---|---|---|---|
| Age | Mean (SD) | 49 (15.22) | 45.8 (13.21) | 0.167 |
| | Median (IQR) | 50 (35, 61) | 47 (34.5, 58.5) | |
| | Range | 18–86 | 22–67 | |
| | 18–55 | 123 (56.9%) | 28 (65.1%) | 0.410 |
| | 55+ | 93 (43.1%) | 15 (34.9%) | |
| Sex assigned at birth | **Female** | **108 (50%)** | **7 (16.3%)** | **<0.001** |
| | **Male** | **108 (50%)** | **36 (83.7%)** | |
| BMI | Mean (SD) | 29.3 (6.04) | 29.7 (6.87) | 0.767 |
| | Median (IQR) | 28.3 (24.7, 32) | 28.7 (24.6, 32.6) | |
| | Range | 15.6–55 | 18.9–49.1 | |
| | <30 | 129 (59.7%) | 23 (53.5%) | 0.556 |
| | ≥30 | 87 (40.3%) | 20 (46.5%) | |
| Race/Ethnicity | **White—non-Hispanic** | **66 (30.6%)** | **8 (18.6%)** | **0.004** |
| | **Black—non-Hispanic** | **22 (10.2%)** | **13 (30.2%)** | |
| | **Hispanic—Latino/a** | **120 (55.6%)** | **20 (46.5%)** | |
| | Other | 8 (3.7%) | 2 (4.7%) | |
| Days from COVID-19 onset to enrollment | Mean (SD) | 51.2 (19.29) | 54.5 (22.7) | 0.372 |
| | Median (IQR) | 50 (36, 66) | 56 (35.5, 68) | |
| | Range | 13–120 | 16–131 | |
| | <28 | 22 (10.2%) | 3 (7%) | 0.116 |
| | 28–<42 | 47 (21.8%) | 12 (27.9%) | |
| | 42–<56 | 59 (27.3%) | 5 (11.6%) | |
| | 56+ | 88 (40.7%) | 23 (53.5%) | |
| COVID-19 severity at enrollment | Asymptomatic | 50 (23.1%) | 9 (20.9%) | 0.862 |
| | Symptomatic | 85 (39.4%) | 16 (37.2%) | |
| | Hospitalized | 81 (37.5%) | 18 (41.9%) | |
| Currently smoke cigarettes or marijuana | **N (%)** | **19 (8.8%)** | **13 (30.2%)** | **<0.001** |
| Ever smoked cigarettes or marijuana | N (%) | 90 (41.7%) | 23 (53.5%) | 0.208 |
| Hypertension | N (%) | 62 (28.7%) | 12 (27.9%) | 1.000 |
| COPD/emphysema/asthma | N (%) | 20 (9.3%) | 5 (11.6%) | 0.843 |
| Diabetes | N (%) | 33 (15.3%) | 5 (11.6%) | 0.703 |
| Prolonged viral shedding | N (%) | 34 (15.7%) | 2 (4.7%) | 0.093 |
| Region | Peru | 117 (91.4%) | 11 (8.6%) | 0.055 |
| | USA | 106 (80.9%) | 25 (19.1%) | |

Significant differences (p values < 0.05) are indicated in bold.

also humoral (four endemic-CoV-specific and one SARS-CoV-2-specific). The response magnitudes of the four endemic-CoV-specific markers (IgG1 to NL63 RBD, IgG1 to OC43 RBD, IgG3 to 229E RBD, and IgG3 to OC43 RBD) were significantly higher in PLWH than PWOH, whereas SARS-CoV-2-specific IgG binding to Spike-transfected cells was significantly lower in PLWH compared to PWOH (Table 2).

**Subsets of immune markers separate PLWH from PWOH**

Because we observed differences in immune profiles among PWOH and PLWH, we hypothesized that a subset of the readouts could distinguish PLWH from PWOH. We used a recursive feature-elimination procedure to down-select immune markers from the 56 readouts and cross-validated super learning to identify combinations of markers that could best classify PLWH from PWOH. Among all participants, regardless of COVID-19 severity, we identified a subset of 12 immune markers that was best able to distinguish PLWH from PWOH, achieving a CV-AUC (cross-validated area under the receiver operating characteristic curve) of 0.756, and a subset

of 7 of the 12 immune markers with a similar CV-AUC of 0.754 (Supplementary Table 7, Supplementary Fig. 3a, b).

No analyses restricted to participants who recovered from asymptomatic infection were carried out because the number of PLWH in this group was low. However, when restricting to participants who recovered from symptomatic outpatient COVID-19, the analysis identified a subset of seven markers potentially predictive of HIV status (Supplementary Table 8). These markers classified PLWH and PWOH with a CV-AUC ranging from 0.48 to 0.69 when evaluated individually and with a CV-AUC of 0.81 when evaluated together (Fig. 2a). To further evaluate these seven markers, we used cross-validated super learning applied to all combinations of the seven markers and identified a subset of three markers (Spike-specific total B cell and memory IgG B cell frequencies and magnitude of endemic OC43-RBD-specific IgG1 binding antibodies) that was best able to classify PLWH from PWOH (CV-AUC = 0.85, 95% CI: 0.74–0.97) (Supplementary Table 8, Supplementary Data 4; Fig. 2b). IgG1 binding antibody response magnitudes to OC43 RBD were significantly higher in PLWH compared to

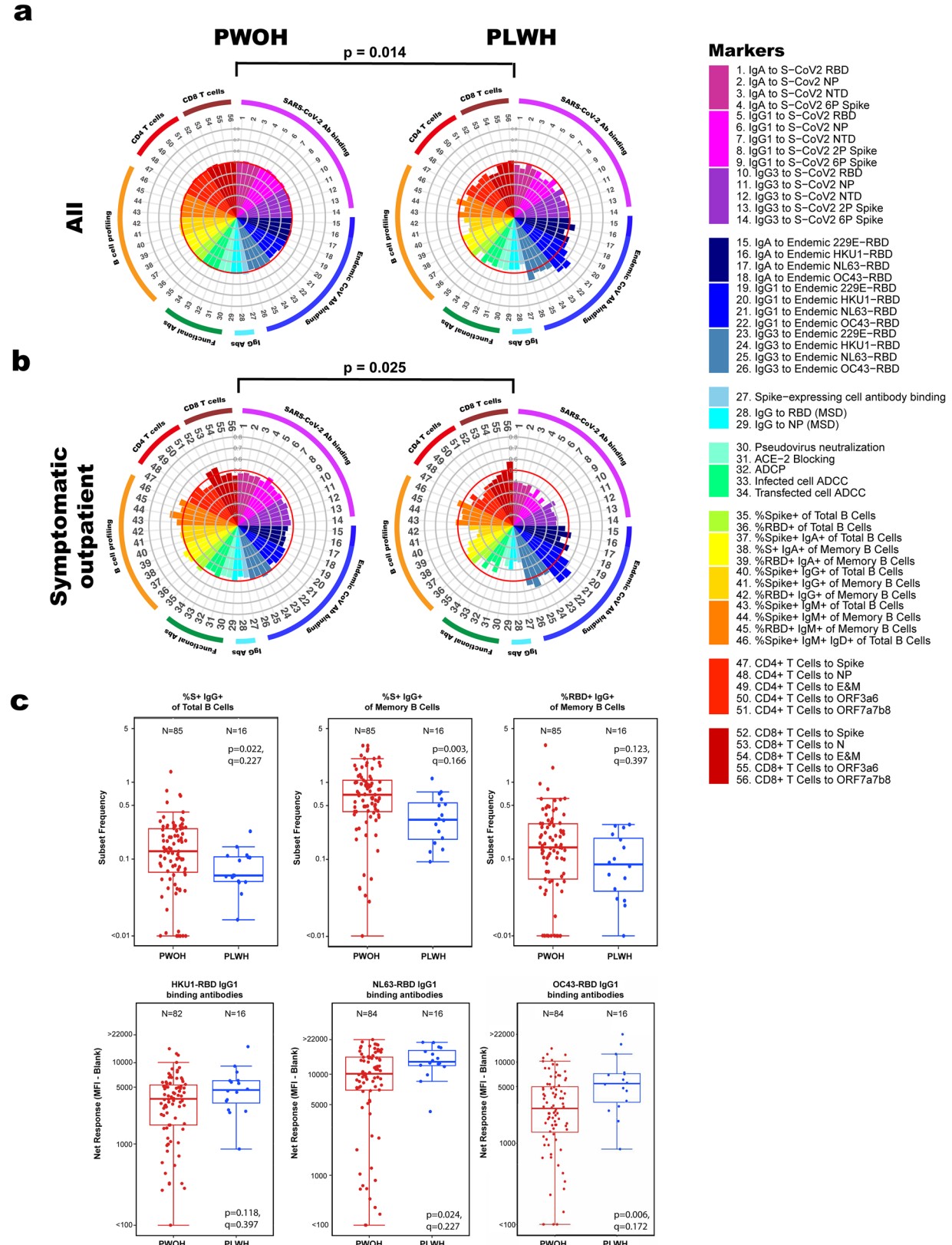

PWOH (Geometric Mean Ratio, GMR: 2.84, 95% CI: 1.36, 5.94) (Fig. 2c; Table 2) whereas Spike-specific total B cell and IgG memory B cell frequencies were lower but not significant (GMR: 0.89, 95% CI: 0.6, 1.31 and GMR: 0.83, 95% CI: 0.37, 1.86, respectively) (Fig. 2c, Supplementary Table 6).

Performing the same classification analysis in participants who recovered from hospitalization yielded five immune markers that were able to distinguish PLWH from PWOH with a CV-AUC = 0.81; these comprised binding antibodies to SARS-CoV-2 (IgG1 to NP and IgG3 to RBD) and endemic CoVs (IgG1 to NL63 RBD and IgG1 and IgG3 to OC43 RBD)

**Fig. 1 | Global differences in immune markers overall between PWOH and PLWH and among symptomatic outpatient participants.** All participants stratified by HIV status: people without HIV (PWOH; $n = 216$) and people living with HIV (PLWH; $n = 42$) (**a**) and participants recovered from symptomatic outpatient COVID-19 ($n = 85$ for PWOH and $n = 16$ for PLWH) (**b**). Polar plots were constructed for all participants (top) or symptomatic outpatient participants (below). Plots indicate the mean quantile for each immune marker and each group. The size of the wedge depicts the mean quantile for the specific subset of participants, ranging from 0 to 1. The red circle indicates the mean quantile (0.5) of all individuals regardless of their HIV status and SARS-CoV-2 disease severity. *P* value indicates the significance using the test of any difference in the global distribution of all immune markers between PWOH and PLWH among all or symptomatic outpatient

participants using the direction-projection-permutation test. Box plots of select immune markers (markers 40–42: SARS-CoV-2-specific IgG+ B cells and markers 20–22: IgG1 binding to endemic CoVs) exhibiting differences between participants without, or living with, HIV recovered from symptomatic outpatient SARS-CoV-2 infection (**c**). The midline of the box denotes the median, and the ends of the box denote the 25th and 75th percentiles among positive responders. The whiskers that extend from the top and bottom of the box extend to the most extreme data points that are no more than 1.5 times the interquartile range (i.e., height of the box) or, if no value meets this criterion, to the data extremes. *P* values were calculated using the Wilcoxon rank sum test, *q* values represent *p* values FDR-adjusted for all 56 markers using the Benjamini and Hochberg method.

**Table 2 | Significant differences in immune markers between PLWH vs PWOH stratified by SARS-CoV-2 infection severity, adjusting for SARS-CoV-2 infection severity, age, sex assigned at birth, currently smoking cigarettes/marijuana, region, and days since SARS-CoV-2 diagnosis at the enrollment**

| Disease status | Immune marker | Magnitude | | | |
| --- | --- | --- | --- | --- | --- |
| | | GMR | 95% CI | *P* value | *Q* value |
| Symptomatic outpatient | IgG1 to SARS-CoV-2 RBD | 0.47 | [0.24, 0.9] | 0.024 | 0.1 |
| | IgG1 to SARS-CoV-2 Nucleoprotein | 0.47 | [0.24, 0.91] | 0.027 | 0.1 |
| | IgG1 to SARS-CoV-2 NTD | 0.31 | [0.12, 0.83] | 0.019 | 0.1 |
| | IgG1 to SARS-CoV-2 6-Proline Spike | 0.25 | [0.12, 0.52] | <0.001 | 0.003 |
| | IgG1 to OC43-RBD | 2.84 | [1.36, 5.94] | 0.006 | 0.033 |
| | MSD IgG to SARS-CoV-2 Spike | 0.43 | [0.21, 0.9] | 0.024 | 0.11 |
| | MSD IgG to SARS-CoV-2 RBD | 0.46 | [0.24, 0.89] | 0.022 | 0.11 |
| | Neutralization (ID50) | 0.53 | [0.29, 0.96] | 0.037 | 0.112 |
| Hospitalized | IgG1 to NL63-RBD | 2.14 | [1.25, 3.68] | 0.006 | 0.033 |
| | IgG1 to OC43-RBD | 2.39 | [1.18, 4.83] | 0.016 | 0.063 |
| | IgG3 to 229E RBD | 2.3 | [1.23, 4.28] | 0.009 | 0.052 |
| | IgG3 to OC43-RBD | 2.54 | [1.39, 4.63] | 0.002 | 0.029 |
| | Spike-expressing cell antibody binding | 0.79 | [0.67, 0.95] | 0.011 | 0.137 |

*GMR* geometric mean ratio.

Significant differences (*p* values < 0.05 and *q* values < 0.2) are indicated in bold.

(Supplementary Table 9; Supplementary Fig. 3a). In this group, PLWH had overall significantly stronger IgG1 and IgG3 responses to endemic CoVs (Table 2) and weaker, but not significantly, IgG3 responses to SARS-CoV-2 RBD and IgG1 responses to the Nucleoprotein (Supplementary Fig. 3c).

**Weaker or negative correlations were observed between SARS-CoV-2-specific markers in PLWH compared to PWOH**

Considering the complexity of immune responses induced by SARS-CoV-2 infection, we explored relationships between the 56 immune markers measured in this study to highlight potential differences in the coordination of immune responses in PWOH and PLWH.

To understand the distribution of correlation coefficients, we plotted the correlation coefficients for each marker pair among PWOH and PLWH as a whole, stratifying the plot by disease severity (Fig. 3a, Supplementary Fig. 4). When assessing the frequency of positive (Spearman rank > 0) and negative correlations (Spearman rank < 0), most correlation coefficients were positive in PWOH (99.09% positive vs 0.91% negative), while 5.58% of correlation coefficients were negative in PLWH (Fig. 3a, left). In addition, more correlation coefficients (1009/1540; 65.5%) fell below the identity line (Fig. 3a, diagonal black lines) among all participants, indicating associations tended to be weaker in PLWH compared to PWOH (Fig. 3a, left), a trend which held true particularly among participants who recovered from symptomatic outpatient COVID-19 (Fig. 3a, right; Supplementary Fig. 4) where 1072/1540 (69.6%) of the correlation pairs were below the identity line. To verify that these observations were not due to the difference in sample sizes between PLWH and PWOH, we generated 10,000 random subsamples from the PWOH groups with the same number of observations

specified as in the respective PLWH groups. In 91.8% and 97.2% of the bootstrap samples, the correlation coefficients were greater in PWOH than PLWH among all participants and among participants who recovered from symptomatic outpatient COVID-19, respectively, compared to 42.7% and 55.7% for participants who recovered from asymptomatic infection and hospitalization, respectively (Supplementary Table 10).

Because participants who recovered from symptomatic outpatient COVID-19 exhibited strong differences in correlation coefficients between PWOH and PLWH overall, we focused on this group and conducted additional analyses. We observed that PLWH displayed weaker correlations between most immune markers, except IgG1 binding antibodies, transfected cell ADCC markers, and CD4 T cells to several proteins (Fig. 3b). Moreover, strong negative correlations observed in PLWH between several immune markers were absent in PWOH, including the frequency of IgM Memory B cells to SARS-CoV-2 RBD and the magnitude of IgG1 and IgG3 responses. Bivariate analyses indicated that the negative correlation was a result of lower IgM Memory B cell frequencies in PLWH compared to PWOH (Fig. 3c). In addition, there were few to no correlations observed between endemic CoV binding antibodies (markers 15–26) and SARS-CoV-2-specific functional antibody responses among participants who recovered from symptomatic outpatient COVID-19, except for IgA binding to OC43 RBD which weakly correlated with SARS-CoV-2-specific neutralization and ADCC and SARS-CoV-2-specific ADCP which moderately correlated with IgG1 binding to endemic 229E and OC43 RBDs in PLWH (Fig. 3b).

Differences in individual correlation coefficients were observed between PWOH and PLWH who had recovered from asymptomatic

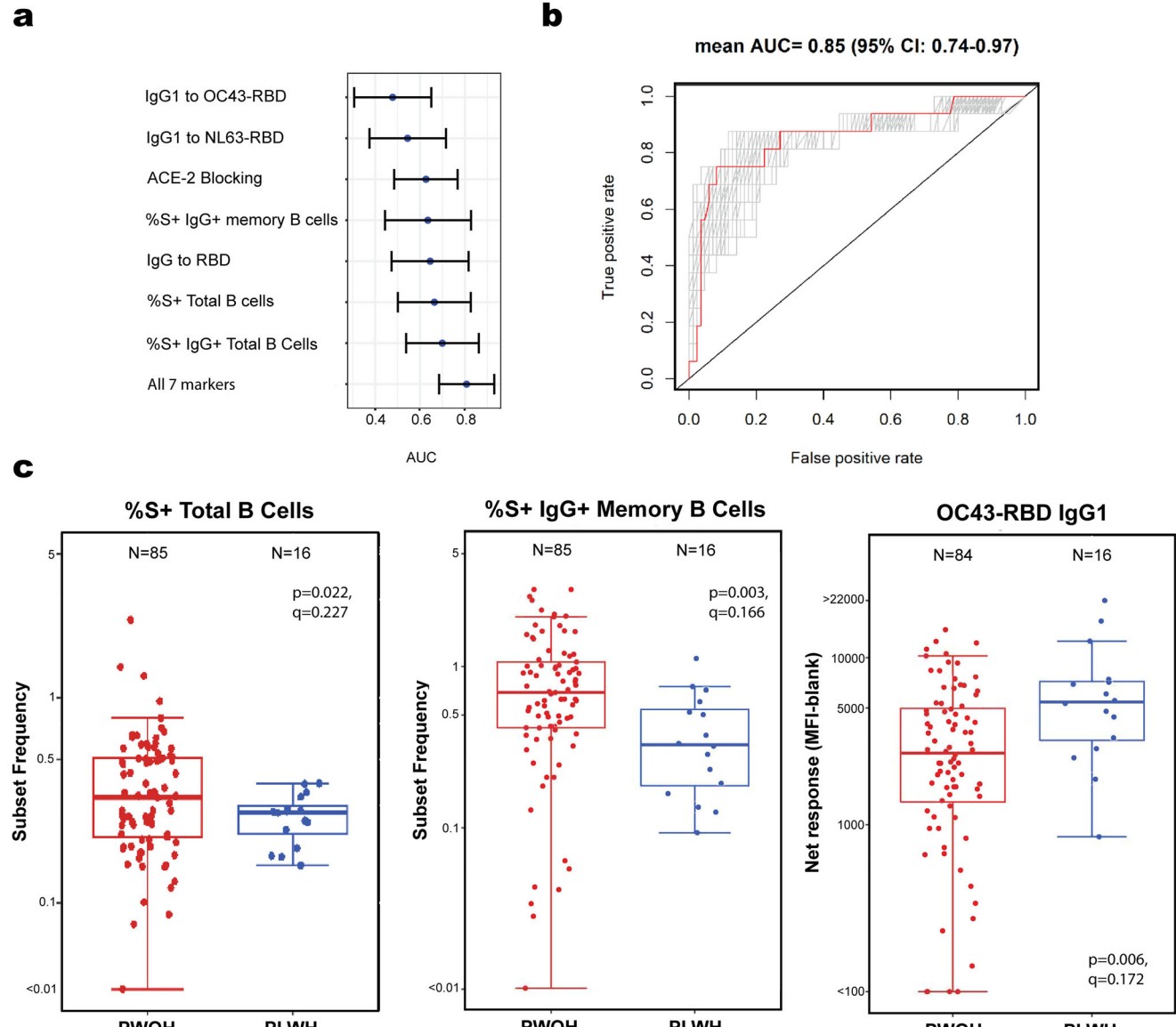

**Fig. 2 | Three immune markers can predict the HIV status of participants recovered from symptomatic outpatient COVID-19 with 85% accuracy.** CV-AUCs with 95% confidence intervals for each of the top seven markers, and all seven markers together, that could classify PLWH recovered from symptomatic outpatient COVID-19 from PWOH using cross-validation super learning (**a**). ROC curves from cross-validation super learning classification of HIV status participants recovering from symptomatic outpatient SARS-CoV-2 infection using the top three immune markers: %S+ Total B cells, %S+ IgG+ Memory B cells, and IgG1 binding to OC43-RBD (**b**). Box plots of immune responses identified as separating PWOH and PLWH recovered from SARS-CoV-2 infection in participants recovered from symptomatic outpatient SARS-CoV-2 infection (**c**). *P* values were calculated using the Wilcoxon rank sum test, *q* values represent *p* values FDR-adjusted for all 56 markers using the Benjamini and Hochberg method.

infection and hospitalization. In participants who recovered from asymptomatic SARS-CoV-2 infection, stronger positive correlations were observed between binding and functional antibody responses in PLWH than PWOH; conversely, negative correlations were observed largely between CD8 T cell, and each of binding antibody, functional antibody, B cell, and CD4 T cell responses in PLWH (Supplementary Fig. 5). Among both PWOH and PLWH recovered from hospitalization, correlations between SARS-CoV-2-specific immune markers were generally weaker than among participants who recovered from asymptomatic infection or symptomatic outpatient SARS-CoV-2 COVID-19.

### SARS-CoV-2 and endemic CoV antibody responses correlate differently depending on disease severity in PLWH

There have been some reports on how the exposure to endemic CoV may influence the anti-SARS-CoV-2 immune responses, but little to no data have been reported for PLWH. Therefore, we sought to understand the relationship between SARS-CoV-2-specific and endemic CoV binding antibody responses when stratifying by HIV status and COVID-19 severity.

Correlations among all PLWH (regardless of disease severity) appeared similar to those in PWOH, whereas stark differences were observed when stratifying by disease severity (Fig. 3b, Supplementary Fig. 5: right panels). Among participants recovered from symptomatic outpatient or hospitalized COVID-19, most correlations between SARS-CoV-2 and endemic CoV antibody binding became weak or negative in PLWH, with the exception of the correlation between SARS-CoV-2-specific IgG3 antibodies and endemic CoV IgG3 antibodies, which became stronger in participants recovered from hospitalization (Supplementary Fig. 5: bottom right panel).

In contrast, SARS-CoV-2 binding antibody responses correlated strongly with several endemic CoV binding antibody magnitudes in PLWH participants recovered from asymptomatic infection (Supplementary Fig. 5,

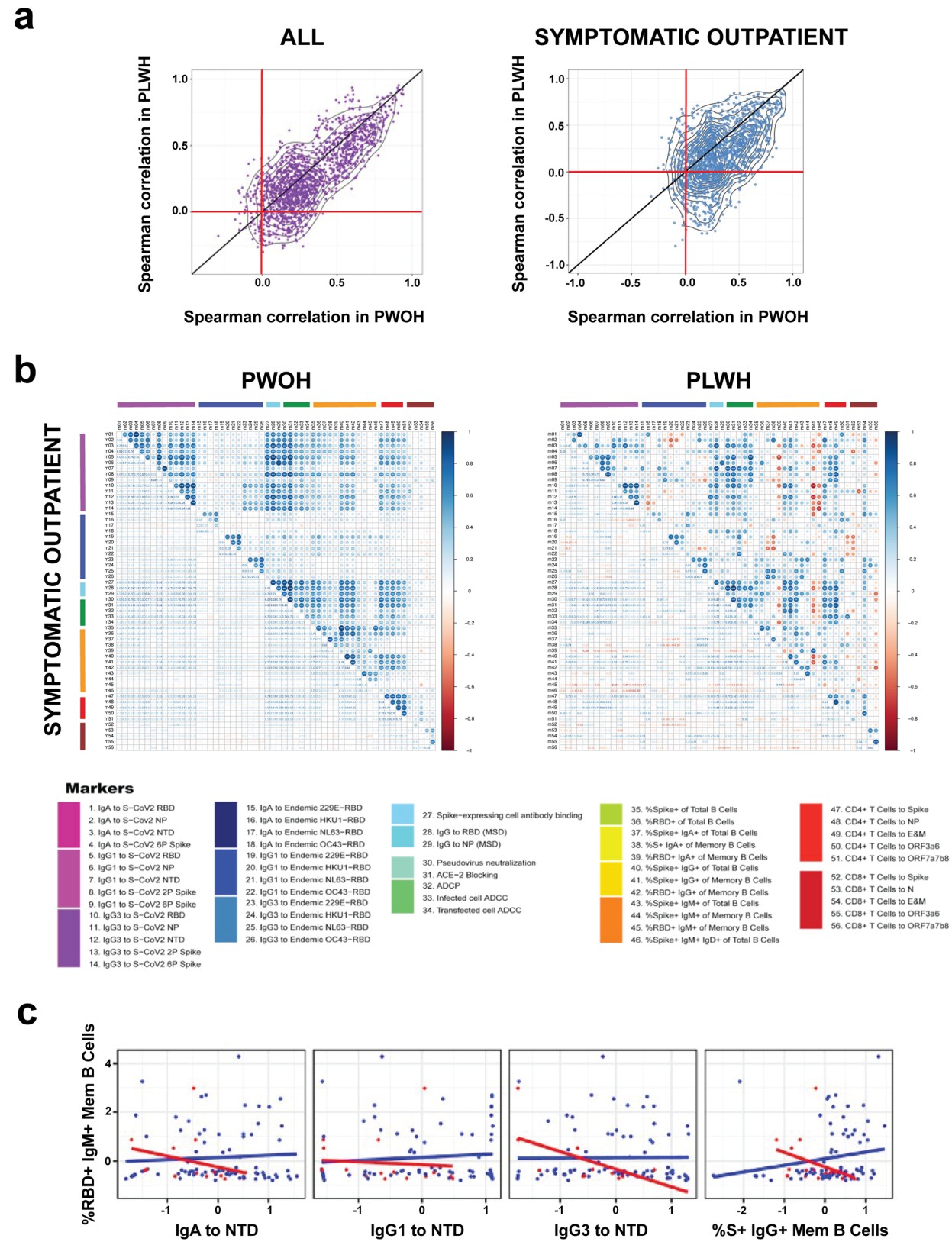

middle right panel). For example, Spearman's rank correlation coefficients for SARS-CoV-2 RBD and Spike-specific IgG3 antibodies (markers 10, 13, and 14) with IgG3 antibodies binding to 229E, HKU1, and NL63 RBDs (markers 23–25) in PLWH recovered from asymptomatic infection were 0.78–0.97 (p values = 0.001–0.017). Interestingly, correlations with IgG3

responses to the beta coronavirus OC43 RBD were lower and not significant (Spearman's rank correlation = 0.4–0.47, p values = 0.29–0.64).

Among PWOH, regardless of COVID-19 severity, only weak correlations were observed between SARS-CoV-2-specific and endemic CoV IgA and IgG1 binding antibody magnitudes (Spearman's rank correlation

**Fig. 3 | Differential distribution and ranges of correlation coefficients between PWOH and PLWH.** Contour plots comparing correlation coefficients in PLWH on the *y*-axis with correlation coefficients in PWOH on the *x*-axis among all participants and symptomatic outpatient participants (**a**). Lines for Spearman correlation = 0 are shown in red. Each point on the contour plot represents a correlation between two immune markers and the identity line (correlation coefficient in PWOH is the same as correlation coefficient in PLWH) is shown in black. Correlation plots of 44 SARS-CoV-2 and 14 endemic CoV immune markers (**b**). Spearman correlations are shown by rho values in the lower triangle and circles in the upper triangle, colored in blue for positive correlation and red for negative correlation; darker colored and larger-sized circles indicate stronger correlations. Significance values are indicated by asterisks in the circles in the upper triangle: *$p$ value ≤ 0.05, **$p$ value ≤ 0.01, and ***$p$ value ≤ 0.001. The significance of Spearman correlation was tested using the exact two-sided test. Scatter plots indicating individual correlation points between %RBD+ IgM+ memory B cells and IgA to NTD, IgG1 to NTD, IgG3 to NTD, and %S+ IgG+ memory B cells (**c**) in PWOH (blue) and PLWH (red). All: $n = 216$ for PWOH, $n = 43$ for PLWH, and symptomatic outpatient: $n = 85$ for PWOH, $n = 16$ for PLWH.

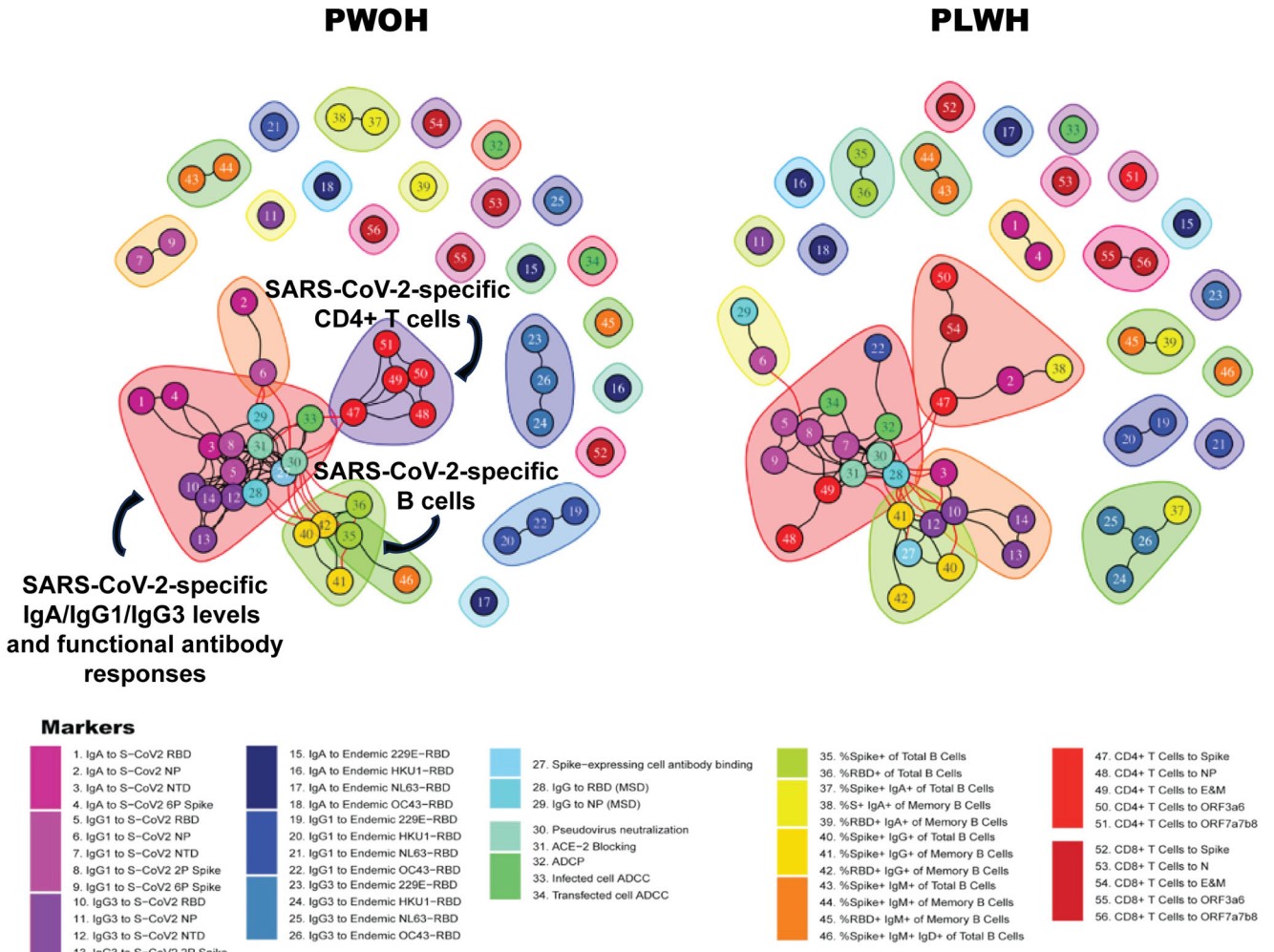

**Fig. 4 | Cluster network analysis of immune markers in PLWH recovered from symptomatic outpatient COVID-19 reveals a less coordinated response than those of PWOH.** Edges between two markers indicate an associated correlation coefficient, which is significant (FDR < 10%) and greater than 0.70 in absolute value based on the exact two-sided test and the Benjamini and Hochberg method. Immune markers are clustered based on edge betweenness. Nodes represent immune markers colored according to marker type. PWOH $n = 85$ and PLWH $n = 16$.

coefficients −0.19–0.49). SARS-CoV-2-specific IgG3 binding antibodies exhibited stronger correlations with endemic CoV IgG3 binding antibodies. Of note, Spearman's rank correlation coefficients for SARS-CoV-2 RBD and Spike-specific IgG3 antibodies (markers 10, 13, and 14) with IgG3 antibodies binding to all endemic RBDs (markers 23–26) in PWOH recovered from asymptomatic infection were 0.53–0.70, with all *p* values < 0.001 (Supplementary Fig. 5: left panels).

### Immune responses are less coordinated in PLWH recovered from symptomatic outpatient COVID-19

We next explored coordination between humoral, T- and B cell responses by building correlation graphs (networks), restricting edges to strong correlations (defined as Spearman's rank correlation with an absolute value >0.7

and significant at the 10% FDR threshold) between all 56 immune markers. Immune markers were clustered based on edge betweenness to identify subgroups of markers that tended to be more closely connected to each other, with each subgroup potentially identifying distinct biological pathways or functions. We focused on participants who recovered from symptomatic outpatient COVID-19 due to the observation that the responses in this group indicated significant differences in PLWH compared to PWOH.

The correlation network for PWOH who recovered from symptomatic outpatient COVID-19 included a large, central cluster encompassing SARS-CoV-2-specific binding antibody responses (IgA, IgG1 and IgG3; purples) and multiple functional antibody measurements (e.g., neutralization, ACE2 blocking and ADCC; blue and green) connected to separate clusters of

**Table 3 | Measures of networks among 56 immune markers for PLWH and PWOH overall and stratified by COVID-19 severity, restricted to significant (FDR < 0.1) and highly correlated (Spearman correlation coefficient > 0.7 or < −0.7) edges or unrestricted**

| | Measure | HIV group | Statistic | All participants (43 PLWH, 216 PWOH) | Asymptomatic (9 PLWH, 50 PWOH) | Symptomatic outpatient (16 PLWH, 85 PWOH) | Hospitalized (18 PLWH, 81 PWOH) |
|---|---|---|---|---|---|---|---|
| Restricted analysis | Size | PLWH | | 88 | 80 | 69 | 65 |
| | | PWOH | | 123 | 123 | 96 | 43 |
| | | 10,000 subsamples from PWOH | Mean (SD) | 144 (46) | 74 (77) | 160 (62) | 75 (33) |
| | | | Median (IQR) | 138 (110,173) | 50 (14,105) | 151 (114,196) | 67 (53,88) |
| | | | Range (Min, Max) | (41,336) | (0,565) | (18,500) | (19,349) |
| | | | **Prob(PWOH > PLWH)** | **0.907** | **0.337** | **0.974** | **0.542** |
| Unrestricted analysis | Size | PLWH | | 949 | 599 | 147 | 662 |
| | | PWOH | | 1353 | 867 | 1072 | 1105 |
| | | 10,000 subsamples from PWOH | Mean (SD) | 1052 (142) | 433 (200) | 677 (207) | 573 (255) |
| | | | Median (IQR) | 1066 (963, 1154) | 441 (298, 575) | 678 (533, 822) | 564 (377, 765) |
| | | | Range (Min, Max) | (450, 1413) | (0, 1114) | (0, 1308) | (0, 1321) |
| | | | **Prob(PWOH > PLWH)** | **0.774** | **0.212** | **0.995** | **0.374** |
| | Average degrees | PLWH | | 34 | 21 | 5 | 24 |
| | | PWOH | | 48 | 31 | 38 | 39 |
| | | 10,000 subsamples from PWOH | Mean (SD) | 38 (5) | 15 (7) | 24 (7) | 20 (9) |
| | | | Median (IQR) | 38 (34, 41) | 16 (11, 21) | 24 (19, 29) | 20 (13, 27) |
| | | | Range (Min, Max) | (16, 50) | (0, 40) | (0, 47) | (0, 47) |
| | | | **Prob(PWOH > PLWH)** | **0.774** | **0.212** | **0.995** | **0.374** |

Total number of possible pairs among 56 immune markers is 1540.
Probability (PWOH > PLWH) for network statistics is indicated in bold.
Measures used were Size (the number of non-zero edges), Average Degree (the number of non-zero edges per marker with the rest of 55 markers), and summary statistics of these measures for 10,000 PWOH subsets that were subsampled from PWOH with the same sample size as PLWH.
*SD* standard deviation; *IQR* Interquartile range.

SARS-CoV-2-specific CD4 T cells (red) by Spike-specific CD4 T cells (marker 47), and IgG+ B cells (green, yellow) (Fig. 4, left panel).

In contrast, networks of immune markers in PLWH who recovered from symptomatic outpatient COVID-19 exhibited less clearly defined and organized immune responses: the clusters contained similar immune markers to PWOH recovered from symptomatic outpatient COVID-19, but network edges were not as frequently observed, resulting in a sparser network (Fig. 4, right panel). Instead of the organized clustering observed in PWOH, IgA and IgG3 binding antibodies were no longer contained in the main cluster of SARS-CoV-2-specific binding and functional antibodies; rather, IgG1 binding to OC43 (marker 22, dark blue) was present in this cluster. In addition, CD4 T cell responses were present in both the main cluster as well as a separate cluster, which consisted of a mixed set of immune markers, including CD8 T cell responses to Envelope and Membrane proteins, IgA to the SARS-CoV-2 NP, and Spike+ IgA+ memory B cells.

To quantify the differences in immune marker coordination between PLWH and PWOH, we first assessed the network sizes. Network size was first defined as the number of significant edges with strong correlations, as outlined above. For participants who recovered from symptomatic outpatient COVID-19, the network size was 69 in PLWH and 96 in PWOH (Table 3). Given the disparity in sample sizes between the two groups, we generated 10,000 random subsamples from the PWOH group, each matched in size to the PLWH group. We then compared the network sizes of these subsamples to those of PLWH. In 97.4% of the 10,000 subsamples, the network size in PWOH (median size = 151, IQR: 114–196) was larger than that in PLWH.

Next, we conducted an analysis without restricting for significance or correlation strength and measured the network size (i.e., the number of non-zero correlations) and average degree (i.e., the average number of non-zero correlations per marker with the remaining 55 markers). This analysis yielded results consistent with the initial findings, with a median network size of 678 (IQR: 533–822) and an average degree of 24 (IQR: 19–29) in

10,000 PWOH subsamples compared to 147 and 5, respectively, in PLWH. In 97.2% of the 10,000 random subsamples, PWOH had a larger network size and average degree than PLWH (Table 3).

These patterns between PWOH and PLWH were not observed among participants who recovered from asymptomatic infection or hospitalization. Among those who recovered from asymptomatic infection, while PWOH had more significant and strongly correlated edges than PLWH (123 vs. 80), only 33.7% of the 10,000 random subsamples (median: 50, IQR: 14–105) showed larger network sizes and average degrees in PWOH than PLWH. This percentage decreased to 21.2% when the analysis was unrestricted (Supplementary Fig. 6, Table 3). Among participants who recovered from hospitalization, PLWH had more significant and highly correlated edges (65 vs. 43 in PWOH). Subsampling resulted in 54.2% of the 10,000 PWOH random subsamples (37.4% in the unrestricted analysis) having larger network sizes and average degrees than PLWH (Supplementary Fig. 6, Table 3).

These findings quantitatively suggest that interactions among key immune response components may be weakened, and immune organization may be reduced in PLWH who recovered from symptomatic outpatient COVID-19 but not in those recovered from asymptomatic infection or hospitalization.

## Discussion

The SARS-CoV-2 pandemic has had a particularly devastating impact on individuals with pre-existing medical conditions, including PLWH. In this study, we applied a systems immunology approach to further understand how immune responses to SARS-CoV-2 in PWOH and ART-treated PLWH differed in the convalescent phase after infection, and how disease severity affected these responses. In individuals recovered from asymptomatic SARS-CoV-2 infection and hospitalization from COVID-19, similar anti-SARS-CoV-2 immune profiles were observed across HIV status. The similar clinical characteristics of the asymptomatic and hospitalized groups

may well explain this, especially considering hospitalization criteria were not well established early in the pandemic.

In comparison, PLWH recovered from symptomatic outpatient COVID-19 exhibited lower humoral and B cell responses to SARS-CoV-2 compared to PWOH. The differences in the profiles of immune responses between PLWH and PWOH recovered from symptomatic outpatient COVID-19 suggest humoral and B cell immune responses elicited against SARS-CoV-2 were either initially lower or shorter-lived in PLWH than in PWOH. Surprisingly, PLWH recovered from both symptomatic outpatient and inpatient COVID-19, but not PLWH recovered from asymptomatic infection, also displayed higher anti-endemic CoV antibody responses than PWOH, indicating that more severe COVID-19 likely raised the levels of pre-existing anti-endemic antibodies.

We did not detect significant differences between PWOH and PLWH who had recovered from asymptomatic infection[40], though the power to detect differences in this group might be limited by a small number of PLWH who experienced asymptomatic infection in the study cohort ($n = 9$). In contrast, several significant differences were observed among participants who had recovered from symptomatic outpatient COVID-19. Differences in SARS-CoV-2-specific binding antibody responses among this cohort have already been reported[17]. In this paper, we have identified additional SARS-CoV-2-specific humoral and cellular immune responses and endemic CoV binding antibody responses that significantly differed between PLWH and PWOH who recovered from symptomatic outpatient COVID-19 and hospitalization due to COVID-19, respectively.

Many of the biomarkers that were down-selected to distinguish PLWH from PWOH in the classification analysis were the result of higher magnitudes of endemic CoV binding antibodies observed in PLWH recovered from symptomatic outpatient and inpatient COVID-19. Importantly, this effect was not observed in PLWH recovered from asymptomatic infection, suggesting that this observation is not solely due to PLWH having a higher susceptibility to endemic CoV infections.

Because these anti-endemic CoV antibodies are specific for the RBD, which has limited conservation among CoVs, and most cross-reactive antibodies observed to date have targeted the Spike S2 subunit[41], the difference in antibody levels likely lies in memory B cell regulation and not in the generation of cross-reactive anti-SARS-CoV-2 antibodies. Aguilar-Bretones et al. observed that, among a cohort of patients hospitalized with severe COVID-19, endemic CoV-specific B cells with limited SARS-CoV-2 cross-reactivity are boosted, resulting in higher levels of endemic CoV antibodies[42]. It is possible that a similar mechanism is resulting in a higher magnitude of endemic CoV antibodies in PLWH recovered from symptomatic outpatient and inpatient COVID-19. Disproportionate boosting of endemic CoV antibody responses is supported by the observation that most of the other immune markers observed to be different between PWOH and PLWH are related to binding antibodies targeting SARS-CoV-2 and differences in B cell population frequencies. Perturbations in B cell function have been observed among PLWH, even in the presence of effective ART[43–45]. In addition, memory B cells in PLWH have demonstrated reduced germinal center activity in the context of COVID-19[46], consistent with a dysregulation that may play a role in the more severe COVID-19 often experienced in PLWH[3–9].

These differences raise important questions about immune responses generated in populations with a high incidence of HIV. Studies of immune responses in vaccinated PLWH in South Africa have resulted in diverging results: vaccination with ChAdOx1 nCoV-19 resulted in comparable magnitudes of binding and neutralizing antibodies among PWOH and PLWH[47], while PLWH exhibited attenuated humoral immune responses and more breakthrough infections when vaccinated with the Novavax NVX-CoV2373 vaccine[48,49]. There is a paucity of data on the effectiveness of mRNA vaccines among PLWH in southern Africa, though it will soon be supplemented by the ongoing CoVPN 3008 COVID-19 mRNA vaccine study (NCT05168813). Importantly, in the context of this ongoing clinical trial, we observed significantly lower neutralization titers, which were correlated with protection in the initial mRNA-1273 COVID-19 vaccine efficacy trial[50], in PLWH recovered from symptomatic outpatient COVID-19, suggesting these responses waned faster in this subgroup than in PWOH.

In addition to the lower magnitude SARS-CoV-2-specific antibody responses observed in PLWH recovered from symptomatic outpatient COVID-19, the correlation networks indicated a significantly less-coordinated immune response against SARS-CoV-2 in PLWH: while antibody binding, functional antibody responses, B cell and CD4+ T cell magnitudes all closely correlated in PWOH, there were fewer nodes within clusters of immune markers in PLWH, with both IgA and IgG3 binding antibodies no longer represented in the main cluster. These qualitative differences in immune coordination were complemented by significantly lower metrics in both the size and average degree of immune marker correlations, as well as the number of significant highly correlated pairs, in PLWH recovered from symptomatic outpatient COVID-19 compared to the PWOH counterpart. These findings were not replicated among participants recovered from asymptomatic infection or hospitalization from COVID-19.

We hypothesize that the lack of differences between PLWH and PWOH among participants who recovered from different disease severity levels in our study may be due to either low viral loads (among participants recovered from asymptomatic infection) resulting in subtle differences in immune coordination that are difficult to distinguish between PLWH and PWOH, or immune activation being so high among participants who recovered from hospitalization that compensatory mechanisms are activated in PLWH, resulting in similar responses between both PLWH and PWOH. Motsoeneng et al. found that different humoral immune markers correlated more strongly among PLWH recovered from hospitalization than the PWOH counterpart[51]. We found similar differences, albeit not significantly, among participants who recovered from hospitalization in this cohort. In addition, Alrubayyi et al. observed low responses among both PWOH and PLWH recovered from mild COVID-19, similar to our study[40]. However, to our knowledge, only this study and another one that assessed humoral immune responses in this cohort[17] have investigated differences between PWOH and PLWH recovered from different levels of COVID-19 disease severity. Consequently, further studies are needed to explore these observations in more detail and to validate them in larger cohorts.

Lastly, we acknowledge that one limitation of this study is its cross-sectional nature, and therefore, the lack of kinetic data and the inability to assign causation of clinical outcomes directly to the immune responses we observed. Another limitation of our study is that feature selection and model evaluation were conducted using the same dataset, which may introduce bias and limit the generalizability of the findings. While we employed cross-validation techniques to internally assess the performance of the model, this does not supplant the need for external validation using an independent dataset. Future research should aim to validate these results in separate cohorts to confirm their robustness and applicability to broader populations. Despite these limitations, this study reveals distinct immunological differences between PWOH and PLWH in response to SARS-CoV-2 infection related to the severity of COVID-19 experienced.

## Conclusion

This study provides insights into the potential relevance of certain immune markers in comparing immune responses between PWOH and PLWH when assessing immune correlates of protection in clinical trials. It also highlights the need to further evaluate the quality of immune responses elicited by vaccination of PLWH and to understand the durability of SARS-CoV-2-specific responses, especially in the context of vaccine-induced protection.

## Data availability

All data underlying the findings of this manuscript are in Supplementary Data 1.

## Code availability

All code used to analyze data and generate figures for this manuscript are available in the following repository DOI: 10.5281/zenodo.15091218[52].

**Article**

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

## Acknowledgements
We thank the HVTN 405/HPTN 1901 study participants and the staff at the 26 participating clinical sites (listed below); DeAnna Tenney, Sir'Tauria Hilliard, Alexander Carnacchi, Judith Lucas, Tara McNair, Michael Archibald, Sheetal Sawant, David Beaumont, Mark Sampson, Angelina Sharak, Kristy Long, LaTonya Williams, Jack Heptinstall for expert technical assistance, data management, specimen management and/or laboratory operations; Marcella Sarzotti Kelsoe for quality assurance oversight QAU; Sarah Mudrak and Valerie Bekker for program management; Statistical Center for HIV/AIDS Research and Prevention (SCHARP) laboratory operations staff for data management; and Drs. Margie Ackerman, Samuel Robinson, and Justin Pollara for expert review of the manuscript. We thank Dr. Neil King of the Institute for Protein Design/ University of Washington for providing the IgG1 mAb CR3022 and the SARS-CoV-2 DB-His constructs; Dr. Leonidas Stamatatos (University of Washington) for the IgG mAbs CV24 and CV30; Nexelis, Dr. Jason McLellan of the University of Texas at Austin and Dr. Barney Graham of the VRC/NIAID/NIH for 2P-stabilized spike (2P) constructs; Dr. McLellan for 6P-stabilized spike construct (6P); and Dr. Peter Kwong (VRC/NIAID/NIH) for the SARS-CoV-2-NTD-AVI construct. Participants were enrolled by the following site investigators/clinical sites: Srilatha Edupuganti (Hope Clinic, Atlanta, GA); Valeria Cantos Lucio (Ponce de Leon, Atlanta, GA); Jason Farley (Johns Hopkins, Baltimore, MD); Paul A. Goepfert (Birmingham, AL); Lindsey R. Baden (Brigham and Women's Hospital, Boston, MA); Kenneth H. Mayer (Fenway Health, Boston, MA); Cynthia Gay (Chapel Hill, NC); Temitope Oyedele (AYAR at CORE, Chicago, IL); Juan Carlos Hinojosa Boyer (Asociación Civil Selva Amazónica, Iquitos, Peru); Javier R. Lama (Barranco, Lima, Peru); Juan Jose Montenegro Idrogo (San Marcos/CITBM, Lima, Peru); Pedro Gonzales (San Miguel, Lima, Peru); Robinson Cabello (Vía Libre, Lima, Peru); Raphael Landovitz (UCLA CARE Center, Los Angeles, CA); Spyros A. Kalams (Vanderbilt, Nashville, TN); Susan Abdalian (Adolescent Trials Unit, New Orleans, LA); Ellen Morrison (Bronx Prevention Center, New York, NY); Yael Hirsch-Moverman (Harlem Prevention Center, New York, NY); Hong Van Tieu (NY Blood Center, New York, NY); Magdalena Sobieszczyk (Physicians and Surgeons, New York, NY); Shobha Swaminathan (New Jersey Medical School, Newark, NJ); Ian Frank (Penn Prevention, Philadelphia, PA); Michael Keefer (University of Rochester, Rochester, NY); Susan P. Buchbinder (Bridge HIV, San Francisco, CA); M. Juliana McElrath (Seattle Vaccine Trials Unit, Seattle, WA); and Manya Magnus (George Washington, Washington, DC). This work was supported by the National Institutes of Health NIAID funded HIV Vaccine Trials Network (HVTN /CoVPN) UM1 AI068618 (Lab); UM1 AI068635 (SDMC); UM1 AI068614 (LOC); UM1 AI148452 (UPenn HIV CTU), UPenn CFAR (P30 AI045008) and Duke CFAR (P30 AI064518), Duke T32 AI141342 (Advanced Immunobiology Training Program for Surgeons), and the Bill & Melinda Gates Foundation (Global Health Vaccine Accelerator Platform (GHVAP) Antibody Dynamics (INV008612)).

## Author contributions
Designing the study—D.M., S.S.L., D.J.S., O.H., G.F.; conducting the experiments—D.M., D.J.S., K.E.S., C.B., B.D., T.K., A.Z., S.S.O., M.S.W., N.E., N.L.Y., X.S., R.S.G.; analyzing data—D.M., S.S.L., X.L., L.Z., J.H., O.H., G.F.; providing reagents—S.K., L.P., J.A.H., L.C., M.J.M., G.D.T.; project administration and supervision—S.K., A.M.S., K.C., A.K.R., S.d.R., M.J.M., G.D.T., O.H., G.F.; writing the manuscript—D.M., S.S.L., X.L., S.K., G.D.T., O.H., G.F.; funding acquisition—L.C., M.J.M., G.D.T. First authorship order was assigned as follows: D.M. was assigned first due to spear-heading design and writing of the study, S.S.L. was assigned second first author due to spear-heading the analysis.

## Competing interests
The authors declare no competing interests.
