## [Transparent Peer Review file · Communications Medicine]

Distinct immune responses in people living with HIV following SARS-CoV-2 recovery

Corresponding Author: Dr Dieter Mielke

Version 0:

Reviewer comments:

Reviewer #1

(Remarks to the Author)

The authors present an interesting project with the hypothesis that the immune response to COVID-19 infection varies according to HIV status. They also use some supervised machine learning tools and network analysis to determine the immune profiles of these populations. Here are a few points I would like to ask the authors to clarify:

- When were the participants included in the study? Were they vaccinated against COVID-19? These questions should be addressed since one can hypothesize that immune response might change according to the subtype of SARS-CoV-2 strains and vaccination status. For instance, the Omicron variant tends to trigger different mechanisms, such as using endocytosis instead of the TMPRSS2-driven pathway, which affects the disease severity and immunologic pathways (<https://doi.org/10.1080/22221751.2021.2023329>). I assume that viral genome sequencing data, or Pango-lineages of viral isolates, might not be collected. Still, at least the timing of the study could give us an understanding of which variants were dominant during the study period.
- The actual meaning of some paper abbreviations, such as 'GMR', should be explicitly provided.
- Interestingly, the global distribution of immune markers differed significantly between PLWH and PWOH among patients who recovered from symptomatic outpatients but not those who recovered from hospitalization. In which aspects are these populations, hospitalized vs. not hospitalized, different? As it is known, hospitalization criteria of patients evolved during the pandemic. While the hospitalization criteria were more conservative in the early periods of the pandemic, strict criteria were set in the following periods, such as low oxygen saturation rate or tachypnea. What were the hospitalization criteria in this study? How do the authors explain the difference in immune profiles between these groups?
- How do the authors explain that the essential immune markers between PLWH and PWOH shown in Table 2 almost differ from those selected as features for the supervised models shown in Table S9?
- Why do the authors choose the cut-off value for a q-value of 0.20? Generally, it is known that an alpha greater than 0.05 is used if there is a tolerance for false positives in the name of discovery. Then, false positives are confirmed as true positives via additional experiments.
- Showing the correlation trends of immune profiles tend to be weaker and disorganized in the PLWH group compared to the PWOH is remarkable and might be emphasized and discussed in more detail with literature findings in the text. However, contrary findings are also available in the literature. For instance, in a recent study, humoral and cellular immune responses were similar between PLWH and PWOH groups after mRNA-based COVID-19 vaccination (<https://doi.org/10.1016/j.ebiom.2023.104661>.)

Reviewer #2

(Remarks to the Author)

In this study, Mielke and colleagues performed a systemic and thorough analysis of humoral and memory B cell responses to SARS-CoV-2 in ART-treated PLWH compared to people without HIV (PWOH). The authors demonstrate that there are significant differences in the immune responses elicited by COVID-19 infection in PLWH compared to PWOH. Overall, this is a clear and concise manuscript. The experiments and analysis are well performed and the conclusions are in line with the data shown, but it is not clear how relevant the results are. There are some points that need to be clarified.

The authors show that HCoV binding antibodies are higher in PLWH recovered from symptomatic COVID-19 compared to PWOH as well as asymptomatic counterparts, but most correlations between SARS-CoV-2 and HCoV antibody binding are weak or negative in symptomatic PLWH, whereas stronger correlations are observed in their asymptomatic counterparts. The authors speculate that the difference in HCoV antibody binding likely lies in memory B cell regulation and not in the generation of cross-reactive anti-SARS-CoV-2 antibodies. The memory B cell responses to HCOVs were not measured, but from existing data analysis it is not clear if there is any association between SARS-CoV-2 specific B cells and HCoV antibody binding in all groups studied.

The authors performed the cluster network analysis for all immune markers that is restricted to strong correlations. It would be interesting to see if there is any association between HCoV antibody binding and SARS-CoV-2 neutralization capacity, ADCC or ADCP?

There are no flow cytometry gating strategies to support SARS-CoV-2-specific memory B cell analysis.

Reviewer #3

(Remarks to the Author)

The authors investigated the humoral and cellular immune responses to SARS-CoV-2 infection between people without HIV and people living with HIV, and found differences in both magnitude and coordination of SARS-CoV-2-specific and anti-endemic CoV immune responses. The manuscript is clearly-written, and the study can help us understand better the immune response of viral infections in immune-compromised individuals.

A few issues:

Figure 1C: there are no statistics. Please calculate false discovery rates (taking into consideration of all the 56 markers listed in figure 1A and B) and put them on figures.

Table 2: explain what GMR is.

Figure 2C: there are no statistics. Add p values, or GMR with CI, etc.

In the section "Subsets of immune markers separate PLWH from PWOH", the authors selected subset of markers that can distinguish PLWH from PWOH. While the authors used the phrase "cross-validated", it is important to note that there was no real "validation" dataset, and the the feature selection and model evaluation were performed using the same dataset. The authors should point out the limitation.

In the network analysis in figure 5, I found it hard to tell the difference between PWOH and PLWH by eye. In both of them, the bottom-left area seemed similarly closely connected. In contrast, regarding asymptomatic and hospitalized groups in figure S7, the difference between PWOH and PLWH is much more obvious by eye. Could the authors come up with a more quantitative measure for how "coordinated" the networks were in this analysis?

If the conclusion in figure 5 still holds using quantitative measures, how to explain that more coordinated responses were observed in PLWH compared to PWOH recovered from asymptomatic infection or hospitalization, while for symptomatic outpatient group it is the opposite?

Regarding the sample size difference between PLWH and PWOH group, the authors noted that "This observation could hypothetically be due to a loss of statistical power because of the lower sample size of PLWH compared to PWOH. However, while less coordinated responses were observed in PLWH recovered from symptomatic outpatient COVID-19, more coordinated responses were observed in PLWH compared to PWOH recovered from asymptomatic infection or hospitalization (Figure S7), suggesting the reduced organization of immune spaces observed in PLWH recovered from symptomatic outpatient COVID-19 cannot be attributed only to a loss of statistical power." While this is a valid argument, other more convincing analysis would be bootstrap/subsampling, for comparison of two groups with unequal sample sizes. Could the authors try it, at least for the spearman correlation analysis (or better, both the spearman correlation and network analyses)?

Version 1:

Reviewer comments:

Reviewer #1

(Remarks to the Author)

The authors have addressed almost all the points highlighted in the first revision with comprehensive and thoughtful explanations. The only point I want to address is the following: While I understand the idea behind the authors' response regarding the definitions of mild and severe illness discussed in Referee 1 Question 3, I would like to emphasize that the definition of severe illness should be presented in detail, especially since it is hypothesized that the lack of difference in overall immune profiles between PLWH and PWOH among participants recovering from different levels of disease severity could be due to low viral loads or excessive immune response. Viral load is not an easily measurable parameter. So, given that the study is directly related to measuring humoral and cellular immunity among these patient groups, there is a need for quantifiable evidence as to why similar anti-SARS-CoV-2 immune profiles are observed between PLWH and PWOH in severe patients. The similar clinical characteristics of the mild and severe disease groups may well explain this picture, especially considering early in the pandemic, hospitalization criteria were not well established. Therefore, the study's results could have been better interpreted if the severity of the patients could have been measured by objective criteria such as hypotension, SO₂ measurement or multi-organ failure.

In conclusion, the manuscript is now more cohesive, and the revision has strengthened the study's theoretical grounding. Overall, I am satisfied with the changes and have no further comments.

Reviewer #2

(Remarks to the Author)

Thank you to the authors for addressing my questions, revising text and adjusting/adding the figures. I endorse the revised manuscript

Reviewer #3

(Remarks to the Author)

The authors have addressed my previous comments, especially the quantitative network analysis. I have no further comments.

We thank the reviewers for taking the time to review this study and provide useful feedback to improve this manuscript. Below we have responded point by point to queries raised and have highlighted changes made to the manuscript to address these points.

Reviewer #1 (Remarks to the Author):

The authors present an interesting project with the hypothesis that the immune response to COVID-19 infection varies according to HIV status. They also use some supervised machine learning tools and network analysis to determine the immune profiles of these populations. Here are a few points I would like to ask the authors to clarify:

1. When were the participants included in the study? Were they vaccinated against COVID-19? These questions should be addressed since one can hypothesize that immune response might change according to the subtype of SARS-CoV-2 strains and vaccination status. For instance, the Omicron variant tends to trigger different mechanisms, such as using endocytosis instead of the TMPRSS2-driven pathway, which affects the disease severity and immunologic pathways (<https://doi.org/10.1080/22221751.2021.2023329>). I assume that viral genome sequencing data, or Pango-lineages of viral isolates, might not be collected. Still, at least the timing of the study could give us an understanding of which variants were dominant during the study period.

The participants were enrolled into the study from May through October 2020, as described in the methods (lines 373-374) “Briefly, participants recovered from SARS-CoV-2 infection were enrolled between May and October 2020 in the HVTN 405/HPTN 1901 observational cohort study (NCT04403880) led by the COVID-19 Prevention Trials Network (CoVPN).”

We have edited the results to include this (line 93-94):” participants enrolled in the HVTN 405/HPTN 1901 trial: a US- and Peru-based study conducted between May and October 2020 (14)”.

Therefore, due to the timing of the study, well before any vaccines were available, no participants were vaccinated.

2. The actual meaning of some paper abbreviations, such as ‘GMR’, should be explicitly provided.

We thank the reviewer for highlighting this missing information. GMR stands for geometric mean ratio. This information has been added to manuscript at the first time we use it (line 184) as well as the legend of tables 2, S1-4, S6, S7, and S8. In addition, ‘Odds Ratio’ (OR) has been added to S1 and S3-7.

3. Interestingly, the global distribution of immune markers differed significantly between PLWH and PWOH among patients who recovered from symptomatic outpatients but not those who recovered from hospitalization. In which aspects are these populations, hospitalized vs. not hospitalized, different? As it is known, hospitalization criteria of patients evolved during the pandemic. While the hospitalization criteria were more conservative in the early periods of the pandemic, strict criteria were set in the following periods, such as low oxygen saturation rate or tachypnea. What were the hospitalization criteria in this study? How do the authors explain the difference in immune profiles between these groups?

a. As correctly indicated by the reviewer, at this early stage in the pandemic, criteria for hospitalization fluctuated both as our understanding of the virus grew, and as hospital capacity varied throughout repeated waves of waxing and waning infection rates. Generally, participant's admission was a physician decision and based on whether or not the participant was determined to be sick enough and in need of hospital care. Consequently, our study did not define hospitalization criteria—that was defined by each individual hospital and participants in this trial were enrolled only after *recovery* (including, in the case of hospitalized participants, after hospital discharge) from acute COVID. However, for any given research site enrolling both hospitalized and non-hospitalized participants, the former typically had more severe disease than the latter; ***thus, across the study cohort overall, hospitalization was a reasonable proxy for disease severity.***

This was also in many ways a factor of the immediacy of this trial's implementation in the earliest months of the pandemic: most clinical research centers could not accommodate, and/or their affiliated institutions did not allow them to have in-person visits with, patients with positive COVID tests or COVID-associated symptoms in their clinics or other institution spaces and only allowed recovered participants to come in. The applicable eligibility criterion for inclusion included on lines 428-431 as following: *“resolution of COVID-19 within 1-8 weeks of enrollment OR, if asymptomatic infection, reports positive SARS-CoV-2 test within 2-10 weeks of enrollment. If they'd been hospitalized, they were not enrolled until after hospital discharge”*.

b. In response to the reviewer's query regarding the differences observed between PLWH and PWOH recovered from symptomatic outpatient and hospitalized COVID-19, we hypothesize that the lack of differences between PLWH and PWOH among participants who recovered from different disease severity levels in our study may be due to either low viral loads (among participants recovered from asymptomatic infection) resulting in subtle differences in immune coordination that are difficult to distinguish between PLWH and PWOH, or immune activation being so high among participants who recovered from hospitalization that compensatory mechanisms are activated in PLWH resulting in similar responses between both PLWH and PWOH. Of note, Motsoeneng et al. (Ref 27) found that different humoral immune markers correlated stronger among PLWH recovered from hospitalization than PWOH. We found similar differences, albeit not significantly, among participants recovered from hospitalization in this cohort. In addition, Alrubayyi et al. (Ref 14) observed low responses among both PWOH and PLWH recovered from mild COVID-19, similar to our study. However, to our knowledge, only this and one other study that assessed immune responses in this cohort have investigated differences between PWOH and PLWH recovered from different levels of COVID-19 disease severity. Consequently, the hypothesis that responses are only different between PWOH and PLWH recovered from symptomatic outpatient COVID-19 needs further testing/support in different cohorts. We have added this to the discussion (lines 396-408).

4. How do the authors explain that the essential immune markers between PLWH and PWOH shown in Table 2 almost differ from those selected as features for the supervised models shown in Table S9?

Thank you for your insightful comment regarding the discrepancy between the essential immune markers in Table 2 and the features selected for the supervised models in Table S9. We appreciate the opportunity to clarify this point.

Firstly, Table 2 shows individual immune markers with significant group-level differences between PLWH and PWOH.

Secondly, due to correlations between markers, not all markers that show differences between PLWH and PWOH individually are needed for a joint classification of PLWH from PWOH. However, some markers that do not show a difference individually can enhance classification accuracy when used jointly with other markers. The feature selection process for supervised models, as described in the Methods lines 948-961, identifies immune markers that collectively improve model accuracy.

As Table S9 shows the selected markers for the classification of PLWH from PWOH for all participants regardless of their COVID-19 severity, Table S10 and Table S12 show the selected markers for the classification of PLWH from PWOH for symptomatic outpatients and hospitalized individuals, respectively.

5. Why do the authors choose the cut-off value for a q-value of 0.20? Generally, it is known that an alpha greater than 0.05 is used if there is a tolerance for false positives in the name of discovery. Then, false positives are confirmed as true positives via additional experiments.

We acknowledge that a different cut-off value for q-value to control false discovery rate (<0.05) has been used in the literature. We used the threshold of 0.20 to generate hypotheses so others can validate the results in different studies and have provided q-values so the readers can select based on their choice of threshold. This has been indicated in the methods (line 658-659): “A q-value of 0.2 was used to be less restrictive and generate hypotheses for future studies.”

Showing the correlation trends of immune profiles tend to be weaker and disorganized in the PLWH group compared to the PWOH is remarkable and might be emphasized and discussed in more detail with literature findings in the text. However, contrary findings are also available in the literature. For instance, in a recent study, humoral and cellular immune responses were similar between PLWH and PWOH groups after mRNA-based COVID-19 vaccination (<https://doi.org/10.1016/j.ebiom.2023.104661>.)

We are aware that diverse findings related to immune profiles of PLWH compared to PWOH have been observed, particularly in the context of infection versus vaccination, and have discussed this in the introduction (lines 65-70). We thank the reviewer for suggesting the above study and have added it to this introductory paragraph (Ref 13).

We have also added discussion to our findings in the context of other literature investigating responses among PWOH and PLWH recovered from different levels of COVID-19 disease severity (lines 416-428). Please see our response to part b of point 3.

Reviewer #2 (Remarks to the Author):

In this study, Mielke and colleagues performed a systemic and thorough analysis of humoral and memory B cell responses to SARS-CoV-2 in ART-treated PLWH compared to people without HIV (PWOH). The authors demonstrate that there are significant differences in the immune responses elicited by COVID-19 infection in PLWH compared to PWOH. Overall, this is a clear and concise manuscript. The experiments and analysis are well performed and the conclusions are in line with the data shown, but it is not clear how relevant the results are. There are some points that need to be clarified.

The authors show that HCoV binding antibodies are higher in PLWH recovered from symptomatic COVID-19 compared to PWOH as well as asymptomatic counterparts, but most correlations between SARS-CoV-2 and HCoV antibody binding are weak or negative in symptomatic PLWH, whereas stronger correlations are observed in their asymptomatic counterparts. The authors speculate that the difference in HCoV antibody binding likely lies in memory B cell regulation and not in the generation of cross-reactive anti-SARS-CoV-2 antibodies. The memory B cell responses to HCoVs were not measured, but from existing data analysis it is not clear if there is any association between SARS-CoV-2 specific B cells and HCoV antibody binding in all groups studied.

1. The authors performed the cluster network analysis for all immune markers that is restricted to strong correlations. It would be interesting to see if there is any association between HCoV antibody binding and SARS-CoV-2 neutralization capacity, ADCC or ADCP?

We agree that assessing the relationship between endemic CoV binding antibodies and SARS-CoV-2-specific functional antibody responses is important. To address this, we have modified the correlation plots to include the endemic binding antibodies (markers 15-26) in Figure 3B for symptomatic outpatients and Figure S5 for all participants, asymptomatic and hospitalized, respectively.

Since the modified Figure 3B and Figure S5 include the endemic binding antibody markers, the original Figure 4 and Figure S6 that show the correlation plots between endemic and SARS-CoV-2 binding antibody markers have been removed.

As shown in the new Figure 3B, there were few to no correlations observed between endemic CoV binding antibodies (markers 15-26) and SARS-CoV-2-specific functional antibody responses among participants recovered from symptomatic outpatient COVID-19, except for IgA binding to OC43 RBD which weakly correlated with SARS-CoV-2-specific neutralization and ADCC and SARS-CoV-2-specific ADCP which moderately correlated with IgG1 binding to endemic 229E and OC42 RBDs in PLWH. We have added this observation to the manuscript (lines 227-233):

“In addition, there were few to no correlations observed between endemic CoV binding antibodies (markers 15-26) and SARS-CoV-2-specific functional antibody responses among participants recovered from symptomatic outpatient COVID-19, except for IgA binding to OC43 RBD which weakly correlated with SARS-CoV-2-specific neutralization and ADCC and SARS-CoV-2-specific ADCP which moderately correlated with IgG1 binding to endemic 229E and OC42 RBDs in PLWH (Figure 3B)”.

2. There are no flow cytometry gating strategies to support SARS-CoV-2-specific memory B cell analysis.

We have added gating strategies for all flow cytometry-based assays to the supplementary information.

Reviewer #3 (Remarks to the Author):

The authors investigated the humoral and cellular immune responses to SARS-CoV-2 infection between people without HIV and people living with HIV, and found differences in both magnitude and coordination of SARS-CoV-2-specific and anti-endemic CoV immune responses. The manuscript is clearly-written, and the study can help us understand better the immune response of viral infections in immune-compromised individuals.

A few issues:

1. Figure 1C: there are no statistics. Please calculate false discovery rates (taking into consideration of all the 56 markers listed in figure 1A and B) and put them on figures.

We calculated p-values of testing differences between PLWH and PWOH using Wilcoxon rank sum test and q-values adjusting for false discovery rate (taking into consideration all the 56 markers listed in figure 1A and B) using Benjamini-Hochberg method and both parameters are included in Figure 1C and its legend.

2. Table 2: explain what GMR is.

GMR stands for geometric mean ratio. See response to reviewer 1 point 2.

3. Figure 2C: there are no statistics. Add p values, or GMR with CI, etc.

Like Figure 1C, we calculated p-values using Wilcoxon rank sum test and q-values adjusting false discovery rate for all 56 immune markers using Benjamini-Hochberg method and included in Figure 2C and its legend.

4. In the section "Subsets of immune markers separate PLWH from PWOH", the authors selected subset of markers that can distinguish PLWH from PWOH. While the authors used the phrase "cross-validated", it is important to note that there was no real "validation" dataset, and the feature selection and model evaluation were performed using the same dataset. The authors should point out the limitation.

We appreciate the reviewer's insightful comment. We acknowledge that our use of the term "cross-validated" referred to internal cross-validation methods (e.g., k-fold cross-validation) performed on the same dataset. However, we recognize that this does not substitute for validation on an independent dataset, which is a limitation of our current study. We have added a statement to the manuscript to highlight this limitation in the discussion (lines 416-421):

“Another key limitation of our study is that the feature selection and model evaluation were conducted on the same dataset, which may introduce bias and limit the generalizability of the findings. While we employed cross-validation techniques to internally assess the performance of the model, this does not replace the need for validation using an independent dataset. Future research should aim to validate these results in separate cohorts to confirm their robustness and applicability to broader populations.”

5. In the network analysis in figure 5, I found it hard to tell the difference between PWOH and PLWH by eye. In both of them, the bottom-left area seemed similarly closely connected. In contrast, regarding asymptomatic and hospitalized groups in figure S7, the difference between PWOH and PLWH is much more obvious by eye. Could the authors come up with a more quantitative measure for how "coordinated" the networks were in this analysis? If the conclusion in figure 5 still holds using quantitative measures, how to explain that more coordinated responses were observed in PLWH compared to PWOH recovered from asymptomatic infection or hospitalization, while for symptomatic outpatient group it is the opposite?

We thank the reviewer for this suggestion. We came up with three quantitative measures: 1) an analysis of the size (the number of pairs (edges)) of the networks, restricted to significant ($FDR < 0.1$) and highly correlated (> 0.7) edges; 2) an unrestricted analysis of the networks using the size; and 3) average degree (average number of non-zero correlations per marker with the remaining 55 markers) of the networks as measures. The method that was used for estimating size and average degree was added in the statistical method section on lines 696-704. The results are now presented in Table 3 and in the result section on lines 297-323.

Based on the new measures of network coordination and random subsampling from the PWOH group, each matched in size to the PLWH group, we found strong evidence (97.4% of subsamples in the restricted analysis and 99.5% in the unrestricted analysis) that PLWH exhibited less coordinated immune responses compared to PWOH among those who recovered from symptomatic outpatient COVID-19. This may reflect the chronic immune activation and exhaustion commonly observed in PLWH, potentially impairing their ability to mount a coordinated immune response during moderate disease severity.

In contrast, among participants who recovered from asymptomatic infection, we observed an opposite trend, with 33.7% of subsamples in the restricted analysis and 21% in the unrestricted analysis showing that PLWH had more coordinated immune responses than PWOH. These findings suggest that any differences in immune coordination between PLWH and PWOH in the asymptomatic group are more subtle and may not be as pronounced as in symptomatic outpatient cases.

For participants who recovered from hospitalization, the approximately equal distribution of network sizes between PLWH and PWOH (with 54.2% and 37.4% of PWOH subsamples showing larger networks in the restricted and unrestricted analyses, respectively) suggests that immune coordination may become more similar between the two groups as disease severity increases. This could be due to the activation of compensatory immune mechanisms in PLWH during severe disease.

We have included these points in the revised manuscript (results lines 298-326 and discussion lines 674-694), highlighting that the evidence for differences in immune coordination is most robust in the symptomatic outpatient group, while only trends or balanced results were observed in the asymptomatic and hospitalized groups. Further studies are needed to explore these observations in more detail and to validate them in larger cohorts.

6. Regarding the sample size difference between PLWH and PWOH group, the authors noted that "This observation could hypothetically be due to a loss of statistical power because of the lower sample size of PLWH compared to PWOH. However, while less coordinated responses were observed in PLWH recovered from symptomatic outpatient COVID-19, more coordinated responses were observed in PLWH compared to PWOH recovered from asymptomatic infection or hospitalization (Figure S7), suggesting the reduced organization of immune spaces observed in PLWH recovered from symptomatic outpatient COVID-19 cannot be attributed only to a loss of statistical power." While this is a valid argument, other more convincing analysis would be bootstrap/subsampling, for comparison of two groups with unequal sample sizes. Could the authors try it, at least for the spearman correlation analysis (or better, both the spearman correlation and network analyses)?

We understand the reviewer's concern and have addressed it by performing similar analyses for subsamples from the PWOH observations having the same sample size as the PLWH counterpart. A total of 10,000 such random subsamples and analyses were carried out. We have summarized the results for the network analysis in Table 3 and the Spearman correlation analysis in Table S13.

We thank the reviewers for their positive assessment of the changes to, and their assistance in improving, this manuscript. Please see our response to Reviewer #1s further comment.

Reviewer #1 (Remarks to the Author):

The authors have addressed almost all the points highlighted in the first revision with comprehensive and thoughtful explanations. The only point I want to address is the following: While I understand the idea behind the authors' response regarding the definitions of mild and severe illness discussed in Referee 1 Question 3, I would like to emphasize that the definition of severe illness should be presented in detail, especially since it is hypothesized that the lack of difference in overall immune profiles between PLWH and PWOH among participants recovering from different levels of disease severity could be due to low viral loads or excessive immune response. Viral load is not an easily measurable parameter. So, given that the study is directly related to measuring humoral and cellular immunity among these patient groups, there is a need for quantifiable evidence as to why similar anti-SARS-CoV-2 immune profiles are observed between PLWH and PWOH in severe patients. The similar clinical characteristics of the mild and severe disease groups may well explain this picture, especially considering early in the pandemic, hospitalization criteria were not well established. Therefore, the study's results could have been better interpreted if the severity of the patients could have been measured by objective criteria such as hypotension, SO₂ measurement or multi-organ failure.

We agree with Reviewer 1 that objective data would be ideal but in the early pandemic setting when this cohort was established (pre-vaccine, hospital systems crashing under the weight of the number of infections, variable adherence to other public health measures, shortages of workforce, personal protection equipment and other hospital resources etc.), a participant's admission was a physician judgement made in the context of the triage that naturally occurred under those circumstance, when limited hospital resources were available and varied across and within regions as COVID-19 prevalence and severity varied over time. Therefore, while hospitalization was a reasonable proxy for disease severity across the study cohort overall and over time, we are faced with the reality that the objective data is unfortunately not consistently available and not necessarily more reliable than the subjective data collected in this study. We believe that further mining of clinical records would introduce additional inconsistencies and variability in the analyses, and not help the reader make a more useful and more consistent interpretation of the clinical data.

We have noted in our discussion that “similar clinical characteristics of the asymptomatic and hospitalized groups may well explain [the similar immune profiles observed between PWOH and PLWH], especially considering hospitalization criteria were not well established early in the pandemic” (lines 753-755), as the reviewer pointed out. We would also like to highlight that a) the same criteria of SARS-CoV-2 infection severity were applied for both PLWH and PWOH and, consequently, differences between these two groups recovered from symptomatic outpatient infection are not likely due to differences in clinical characteristics and b) although the difference in immune profiles between PLWH and PWOH among hospitalized participants is not statistically significant ($p=0.14$), it is in the same trend as the one for the symptomatic outpatient group.